# ARES: Multimodal Adaptive Reasoning via Difficulty-Aware Token-Level Entropy Shaping

**Shuang Chen**[1]*, **Hangyu Guo**[1]*, **Yimeng Ye**[3], **Shijue Huang**[2], **Wenbo Hu**[1], **Jiayu Chen**[2],
**Manyuan Zhang**[4], **Haoxi Li**[2], **Song Guo**[2], **Nanyun Peng**[1]
[1]University of California, Los Angeles    [2]The Hong Kong University of Science and Technology
[3]Columbia University    [4]The Chinese University of Hong Kong

## Abstract

Recent advances in multimodal large reasoning models (MLRMs) have substantially improved their ability to solve complex textual and visual tasks. However, these models tend to *overthink* on simple problems, producing unnecessarily lengthy reasoning traces, while *under-exploring* on challenging ones, leading to missed solutions. To address this imbalance, we propose **ARES**, a unified open-source framework for *adaptive reasoning* that dynamically allocates exploration effort based on task difficulty. Our approach is motivated by two key empirical findings: (i) while single-token entropy is noisy, *high window-entropy (HWE) tokens* (token-level entropies averaged under a sliding window) can reliably capture reasoning-critical moments; and (ii) reducing HWE usage benefits easy problems, while increasing it is essential for solving hard ones. Building on these insights, ARES introduces a two-stage training pipeline. In the *Adaptive Cold-Start* stage, we curate multimodal and textual data paired with reasoning traces of length proportional to problem difficulty, equipping the model with initial difficulty awareness. In the second stage, we develop *Adaptive Entropy Policy Optimization (AEPO)*, which uses HWE tokens as exploration triggers to decide *when to explore*, and a hierarchical entropy reward with dynamic KL control to decide *how much to explore*. Extensive experiments demonstrate that ARES achieves state-of-the-art performance and reasoning efficiency across diverse mathematical, logical, and multimodal benchmarks, while closing the gap to leading commercial systems under significantly lower inference costs. The anonymous code repository is available at https://github.com/shawn0728/ARES.

## 1 Introduction

Leveraging long Chain-of-Thought (CoT) reasoning with reflection, Multimodal Large Reasoning Models (MLRMs) have showcased strong capabilities on both complex textual and visual tasks (Comanici et al., 2025; Guo et al., 2025). While incorporating long CoT reasoning substantially improves performance on complex reasoning tasks, it often causes models to generate unnecessarily lengthy reasoning even for easy tasks (Qu et al., 2025). This substantially increases inference costs and latency, limiting MLRMs' usability in various real-world scenarios.

Recent studies have explored both training-free (Han et al., 2025; Yang et al., 2025b) and training-based strategies (Zhang et al., 2025; Arora & Zanette, 2025; Ling et al., 2025; Tu et al., 2025) to mitigate the issue of verbose long-thought outputs. While these methods alleviate overthinking, they often cause model performance degradation (Huang et al., 2025b). To balance model's efficiency and accuracy, adaptive reasoning mechanisms have become a promising research direction, which dynamically adjust reasoning effort to trade off performance against computational cost. Existing approaches attempt this by either curating cold-start data with varying difficulty (Wang et al., 2025d; Zhang et al., 2025; Yu et al., 2025a) or using difficulty-aware penalties during RL (Huang et al., 2025a; Shen et al., 2025b). However, such methods frequently encourage exploration on difficult

---

*Equal contributions.

problems, leading to unnecessarily verbose reasoning traces, and they still fail to deliver significant performance improvements.

To address these challenges, we seek to answer two foundational questions for adaptive reasoning mechanism: *When should a model be encouraged to explore? How much reasoning effort should be allocated during exploration?* To answer this, the paper presents a detailed preliminary analysis of exploration during the RL stage of MLRMs, specifically investigating how exploration affects the overall model's performance and efficiency. We meticulously design experiments by constructing an easy, medium and hard set for study the trade off of RL exploration cost against accuracy. As illustrated in Figure 1, our findings can be summarized in two key observations. Firstly, high token-level entropy in MLRMs not only captures tokens relevant to reasoning but also often includes noise such as punctuation, formulas, or other superficial elements. By comparison, window entropy, the mean entropy computed over consecutive tokens, more reliably identifies pivotal decision points that guide the reasoning trajectory across multiple potential pathways. Secondly, for easy problems, reducing the number of high window entropy (HWE) tokens shortens the reasoning trace and improves performance. For complex problems, by contrast, increasing the number of HWE tokens promotes more thorough exploration, thereby enhancing the model's ability to solve challenging tasks. These findings highlight HWE tokens as an exploration trigger for adaptive reasoning in MLRMs, improving performance while reducing inference costs.

Motivated by these empirical findings, we propose an adaptive reasoning training approach called ARES, standing for multimodal **A**daptive **R**easoning via difficulty-aware token-level **E**ntropy reward **S**haping. Specifically, first, during the Adaptive Cold-Start stage, we instill an initial difficulty awareness by fine-tuning the model on data where reasoning length is explicitly correlated with problem complexity. Second, we introduce **A**daptive-**E**ntropy **P**olicy **O**ptimization (AEPO), which uses high window-entropy regions to trigger exploration and a difficulty-aware hierarchical reward to control its depth. This dual-stage approach enables ARES to adaptively allocate reasoning effort, rewarding deep exploration on complex problems while penalizing overthinking on simpler ones.

In summary, our main contributions are as follows:

- We empirically demonstrate that high–window-entropy (HWE) tokens can guide MLRMs in exploration and reveal distinct exploration behaviors between easy and hard tasks.

- We introduce ARES with a high-quality cold-start pipeline grounded in our curated ARES-SFT-224K dataset. Together with AEPO's token-level entropy–shaping RL strategy, training on these datasets consistently improves model performance.

- Extensive experiments demonstrate that ARES achieves superior performance and inference efficiency across a wide range of mathematical, general knowledge, textual, and multimodal reasoning benchmarks.

## 2 RELATIONS BETWEEN TOKEN ENTROPY AND REASONING DIFFICULTY

In this section, we show two intriguing findings involving the exploration and effectiveness of MLRMs, paving the way for the proposed ARES in Section 3.

### 2.1 WINDOW ENTROPY SERVES AS A TRIGGER FOR EXPLORATION IN MLRMS

Recent works (Wang et al., 2025b; Zheng et al., 2025) highlight that *high-entropy tokens* often mark the starting points of reasoning forks, where the model must branch into alternative continuations. Such points are critical for determining the depth of reasoning, since they open up new solution paths. However, as discussed in Appendix C, token-level entropy $H_t$ only captures local uncertainty at a single step and is often dominated by short-lived lexical ambiguities (e.g., punctuation, function words). Conversely, discourse markers that indicate major logical transitions (e.g., "but," "however") may appear with very low entropy, despite signaling important shifts in reasoning. Hence, relying solely on single-token entropy is noisy and insufficient for reliably identifying when exploration should be triggered.

From a cognitive perspective, humans rarely experience uncertainty as an instantaneous hesitation on a single word. Instead, when encountering a conceptual bifurcation, an entire segment of reasoning tends to remain unstable. Motivated by this intuition, we aggregate token-level en-

tropies into a sliding-window statistic that captures the persistence of uncertainty across consecutive steps: $\bar{H}_{t:w} = \frac{1}{w}\sum_{\tau=t}^{t+w-1} H_\tau$. This measure highlights regions where the model sustains high uncertainty over multiple tokens, thus providing a smoother and more semantically aligned indicator of reasoning-critical moments. Window entropy therefore reduces the noise of single-token signals while better localizing genuine reasoning bifurcations.

Our empirical analysis supports this intuition. As shown in Figure 5 in Appendix, window entropy achieves consistently higher F1 scores than single-token entropy for detecting reasoning-critical tokens. Moderate windows (4–8 tokens) offer the best trade-off: they smooth local noise while remaining focused enough to capture sustained uncertainty. In contrast, single-token measures are overly sensitive to lexical artifacts, and long windows (16–32 tokens) dilute the signal with low-entropy tokens. These results validate window entropy as a more robust diagnostic of reasoning uncertainty and motivate its role as a central trigger in our Adaptive Exploration Policy Optimization (AEPO) framework (Section 3.2.1).

## 2.2 Entropy–Difficulty Interaction

Exploration should not be uniform across all problem difficulties. Intuitively, for *easy* problems, the model already possesses sufficient knowledge to arrive at the correct answer; excessive high-entropy branching in such cases is likely to lead to unnecessary verbosity or even distract from the correct reasoning path. In contrast, *hard* problems inherently require sustained exploration: reaching the correct answer typically involves probing multiple solution paths, which manifests as longer responses and more high-entropy tokens.

We validate these intuitions by analyzing model generations using window-entropy statistics. As illustrated in Figure 1, exploration interacts strongly with problem difficulty. In the *easy* set, samples with high-entropy counts `below` a batch-adaptive threshold attain higher accuracy while producing shorter responses, indicating that suppressing unnecessary exploration is beneficial. By contrast, in the *hard* set, `above`-threshold samples achieve higher accuracy but at the cost of longer responses, showing that sustained exploration improves success on challenging problems.

Conditioning on correctness further highlights this pattern. For *easy* problems, correct cases contain markedly fewer high-entropy tokens and shorter responses compared to incorrect ones. For *hard* problems, correct cases typically involve longer responses and more high-entropy tokens than incorrect ones, consistent with the need for extended exploratory reasoning. Together, these results establish a clear entropy–difficulty interaction: exploration should be suppressed for easy tasks but encouraged for hard ones.

These results establish a clear *entropy–difficulty interaction*: on easy problems, suppressing exploration reduces verbosity and improves accuracy, while on hard problems, encouraging additional high-entropy reasoning is crucial for success. This motivates our difficulty-aware exploration strategy, where entropy-based triggers are adaptively modulated across difficulty buckets to balance performance and reasoning efficiency. As further justified in Appendix L, we establish theoretically that response length grows approximately linearly with the number of high-entropy tokens, validating entropy as a principled proxy for reasoning effort.

## 3 Method

We aim to endow a multimodal policy with the ability to *adapt* its reasoning depth: produce short answers for easy instances and longer, exploratory chains for hard ones. Our **ARES** method consists of two stages: a cold-start curriculum (AdaCS) that preserves length-controllable modes, followed by a KL-regularized RL stage (AEPO) that couples a token-level *high-entropy window* detector with a *difficulty-aware* KL budget, which is illustrated in Figure 2.

### 3.1 AdaCS: Adaptive ColdStart Fine-tuning

Inspired by Wang et al. (2025d), during the cold-start phase, training with simple problems paired with short chains of thought (CoT) and difficult problems paired with long CoTs effectively enhances the model's difficulty awareness and facilitates the acquisition of key high-window-entropy (HWE) tokens as well as reflective reasoning capabilities. We first curate a high-quality set of RLVR textual

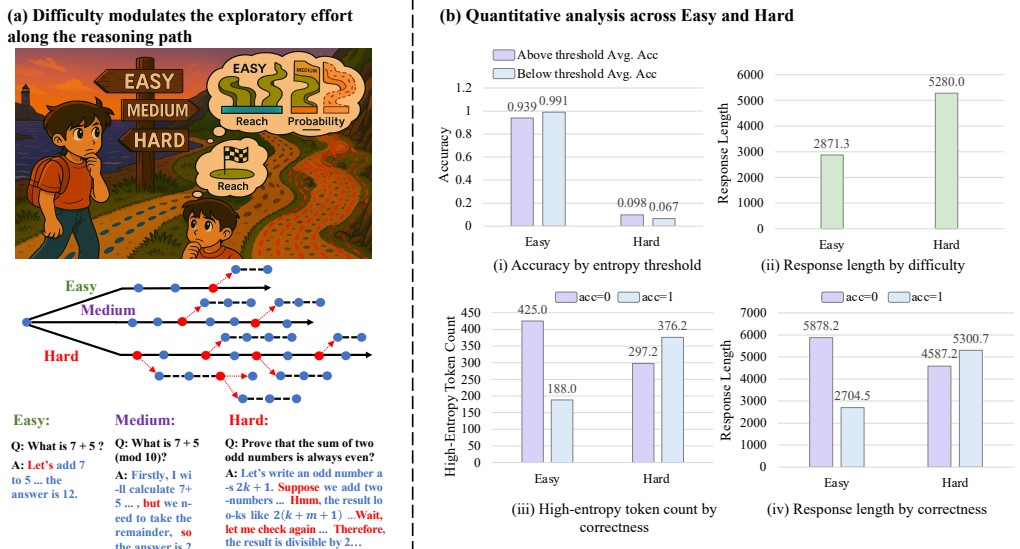

Figure 1: **Entropy–difficulty interaction in exploration.** (a) Conceptual illustration: task difficulty modulates the reasoning trajectory, with easy problems requiring little exploration and hard problems benefiting from deeper branching. (b) Quantitative analysis: (i) for easy tasks, responses below the entropy threshold are both shorter and more accurate; for hard tasks, above-threshold exploration yields higher accuracy; (ii) response length increases significantly with difficulty; (iii) within each difficulty, correct cases use fewer high-entropy tokens for easy problems but more for hard problems; and (iv) correctness further amplifies this trend in response length. Together, these results show that limiting exploration improves efficiency on easy problems, while encouraging additional exploration is crucial for solving difficult ones.

data and multimodal STEM tasks, which serves as the foundation for the cold-start stage. More details regarding dataset sources and preprocessing can be found in Appendix 4.1. Different from prior work (Wang et al., 2025d) that discards samples with a pass rate of 1 and oversamples those with lower pass rates, our methodology is specifically designed to train the model to differentiate problem difficulty and generate responses of corresponding lengths. To achieve this, for each data source, we establish a target response length for every pass rate bracket, which is determined by the median response lengths of the easiest (pass rate = 1) and hardest (pass rate = 0) problems. We then sample responses whose lengths are proximate to this target, while ensuring uniform sampling across all pass rate brackets. This strategy is motivated by the objective of instilling a strong, explicit correlation between perceived problem difficulty and the verbosity of the generated rationale. Formally, for any intermediate pass rate $p \in (0, 1)$, the target response length is defined as $L_{\text{target}}(p) = (1 - p) \cdot L(0) + p \cdot L(1)$, where $L(0)$ and $L(1)$ denote the median token lengths of responses for problems with pass rates of 0 and 1, respectively, for a given data source. This design maximizes response length diversity across pass rates, thereby amplifying the distinction between easy and hard problems and enabling the model to better learn difficulty-aware reasoning strategies.

## 3.2 AEPO: ADAPTIVE-ENTROPY POLICY OPTIMIZATION

To overcome the imbalance between overthinking easy problems and under-exploring difficult ones (Section 2.2), we introduce **Adaptive-Entropy Policy Optimization (AEPO)**. The central idea is to endow the policy with an *adaptive exploration mechanism* that decides both *when to explore* and *how much to explore*, guided by entropy signals. A detailed flowchart of AEPO is presented in Algorithm 1 in the Appendix.

### 3.2.1 WHEN TO EXPLORE.

To operationalize window entropy as an exploration trigger, we introduce a bucket-wise high-entropy threshold. Specifically, for each rollout trajectory, we compute the token-level entropies $\{H_t\}_{t=1}^{T}$ and extract the $95^{\text{th}}$ percentile value as the high-entropy threshold for that sequence. This choice is motivated by recent empirical findings (Wang et al., 2025b), which show that RLVR predominantly reshapes the entropy distribution of tokens in the top $5\%$ percentile range, while low-entropy tokens remain comparatively stable. Hence, the $95^{\text{th}}$ percentile serves as a robust indicator

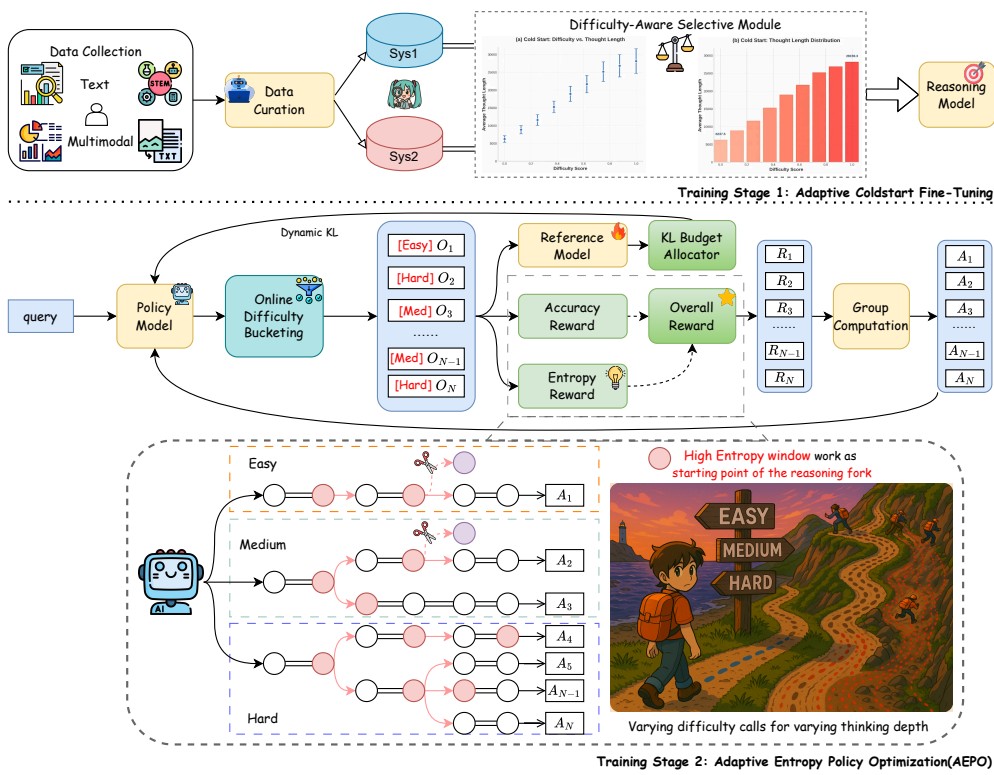

Figure 2: Overall training pipeline of our method. Stage 1 (Adaptive Coldstart Fine-Tuning): difficulty-aware selective data curation and adaptive KL-guided fine-tuning establish a strong initialization across text and multimodal inputs. Stage 2 (Adaptive Entropy Policy Optimization, AEPO): online difficulty bucketing and entropy-aware rollout allocate reasoning depth dynamically, with high-entropy windows serving as branching points for exploration. Together, the two stages enable uncertainty-aware, difficulty-adaptive reasoning for large language models.

of reasoning-critical uncertainty. To reduce noise at the single-sequence level, the thresholds are then averaged across trajectories in a mini-batch to yield a stable batch-level cutoff $\tau_{\text{high}}$:

$$\tau_{\text{high}} = \frac{1}{|\mathcal{D}|} \sum_{y \in \mathcal{D}} \text{Quantile}_{0.95}(\{H_t(y)\}_{t=1}^{|y|}),$$

where $\mathcal{D}$ denotes the set of trajectories in a batch.

During training, this threshold is updated dynamically on a batch-by-batch basis, allowing AEPO to adaptively align the exploration schedule with the evolving entropy distribution. At rollout time, whenever the windowed entropy $\bar{H}_{t:w}$ of a segment exceeds $\tau_{\text{high}}$, the model branches additional trajectories from step $t$. At each high-entropy window, exactly one additional trajectory is generated, unless the total number of generated trajectories has already reached the maximum allowed. In this way, exploration is triggered only at sustained high-uncertainty regions, concentrating computational resources on reasoning-critical moments while avoiding unnecessary branching at stable, low-entropy segments.

### 3.2.2 HOW MUCH TO EXPLORE.

Motivated from our preliminary studies, we next investigate the principle of *how much exploration should be allocated once a reasoning-critical region is detected*. While window entropy serves as a reliable trigger for *when* to explore, the amount of exploration must be carefully controlled to avoid excessive reasoning on simple tasks and insufficient exploration on complex ones. To achieve this balance, AEPO integrates two complementary mechanisms:

**Hierarchical Reward Design.** The goal of AEPO's hierarchical reward is to adaptively regulate *the degree of exploration* once reasoning-critical regions are detected. Building on the online difficulty buckets (Appendix H), we integrate accuracy and entropy-based shaping into a single constraint-driven formulation that requires no additional hyperparameters.

Formally, for a trajectory $y$ with $N_{\text{HE}}$ high-entropy tokens, we define the bucket-dependent target as the batch mean:

$$N_{\text{HE}}^{\text{target}}(d) = \mathbb{E}_{\text{batch}}[N_{\text{HE}} \mid d], \qquad d \in \{\texttt{easy}, \texttt{medium}, \texttt{hard}\},$$

which is updated online at each iteration to track the evolving distribution of exploratory behavior. To regulate deviations from this target, we introduce a closed-form Lagrange multiplier derived from batch statistics:

$$\lambda_d = \max\left(0, \frac{\mathbb{E}_{\text{batch}}[N_{\text{HE}} \mid d] - N_{\text{HE}}^{\text{target}}(d)}{\text{Var}_{\text{batch}}[N_{\text{HE}} \mid d] + \varepsilon}\right).$$

This term automatically scales the strength of entropy shaping without requiring manually tuned weights or learning rates.

Let $\Delta(y; d) = N_{\text{HE}} - N_{\text{HE}}^{\text{target}}(d)$. The shaping direction is defined as:

$$g_d(\Delta) = \begin{cases} \max(0, \Delta), & d = \texttt{easy}, \\ |\Delta|, & d = \texttt{medium}, \\ \max(0, -\Delta), & d = \texttt{hard}. \end{cases}$$

The qualitative effect of $g_d(\Delta)$ differs across difficulty levels. For easy problems, positive deviations ($\Delta > 0$) correspond to unnecessary over-exploration and are penalized. For medium problems, both under- and over-exploration are discouraged, resulting in a symmetric shaping curve around the target. For hard problems, negative deviations ($\Delta < 0$) indicate insufficient exploration and are penalized, thereby encouraging longer and more exploratory reasoning chains. We provide a detailed visualization and discussion of these curves in Appendix J (Figure 7).

Finally, the overall hierarchical reward is given by:

$$R(x, y; d) = R_{\text{acc}}(x, y) - \mathbf{1}[\text{acc}(x, y) = 0] \; \lambda_d \, g_d\big(\Delta(y; d)\big),$$

where hierarchical reward $R(x, y; d)$ unifies the correctness term $R_{\text{acc}}(x, y)$ with a difficulty-aware entropy regularization. Specifically, the entropy penalty is applied only when the response $y$ is incorrect, ensuring that already-correct solutions are not discouraged while encouraging exploration on failed attempts.

This formulation establishes a difficulty-aware intrinsic reward that suppresses unnecessary exploration on easy tasks, stabilizes reasoning depth around a batch-adaptive target on medium tasks, and promotes sustained exploration on hard tasks. Importantly, the entire scheme operates in a closed-form manner based solely on batch-level statistics, thereby achieving adaptive exploration control without introducing additional hyperparameters.

**Dynamic KL Design.** As established in Appendix M, the KL loss provides a valid *thinking budget*, while Appendix F further shows that a KL penalty inflates variance, motivating our exclusive use of the KL loss. To realize adaptive exploration, AEPO employs only a *token–adaptive weight* mechanism:

$$\beta_{i,t} = \beta_d \cdot \rho_t, \qquad \rho_t = \begin{cases} \rho \, (< 1), & \text{if } t \in \mathcal{W}^{\text{valid}}, \\ 1, & \text{otherwise}, \end{cases} \tag{1}$$

where $\beta_d$ is the difficulty–dependent baseline and $\rho_t$ relaxes the KL constraint inside validated high–entropy windows. This simple yet effective design ensures that KL divergence is tightly controlled on easy tokens but adaptively relaxed at reasoning-critical segments, functioning as a token-wise *thinking budget allocator*.

Then we present the surrogate objective of AEPO, which is adapted from the GRPO and DAPO formulations discussed in Appendix D. The objective function is defined as:

$$\mathcal{J}_{\text{AEPO}}(\theta) = \mathbb{E}_{(q,a) \sim \mathcal{D}, \, \{o^i\}_{i=1}^G \sim \pi_{\theta_{\text{old}}}(\cdot | q)} \left[ \frac{1}{G} \sum_{i=1}^G \frac{1}{|o^i|} \sum_{t=1}^{|o^i|} \min\left( r_{i,t}(\theta) \, \tilde{A}_{i,t}, \right. \right.$$

$$\left. \left. \text{clip}\big(r_{i,t}(\theta), 1 - \epsilon_\ell, 1 + \epsilon_h\big) \tilde{A}_{i,t} \right) - \beta_{d(i),t} \, D_{\text{KL}}\big(\pi_\theta(\cdot \mid s_{i,t}) \,\|\, \pi_{\text{ref}}(\cdot \mid s_{i,t})\big) \right], \tag{2}$$

where $r_{i,t}(\theta) = \frac{\pi_\theta(o_t^i|s_{i,t})}{\pi_{\text{ref}}(o_t^i|s_{i,t})}$, $s_{i,t} = (q, o_{<t}^i)$, and $\tilde{A}_{i,t}$ is the task+entropy shaped (token-level) advantage induced by the hierarchical reward (Section 3.2.2).

# 4 EXPERIMENTS

## 4.1 EXPERIMENTAL DETAILS

**Datasets** The training of ARES follows our proposed methodology, utilizing carefully curated datasets for each stage. The initial adaptive cold-start phase employs a dataset of approximately 224K samples, including both high-quality textual data and multimodal STEM tasks (Table 5). We then employ a strong open-source cold-start model, Revisual-R1 (Chen et al., 2025b), to generate eight responses per sample and compute the pass rate. To establish a robust initial policy, textual problems are answered using the powerful DeepSeek-R1 model (DeepSeek-AI et al., 2025), while multimodal reasoning tasks are addressed with Seed-1.5-VL (Guo et al., 2025). The subsequent RLVR stage with AEPO then utilizes the ViRL39K dataset (Wang et al., 2025a), a collection of verifiable question–answer pairs derived from existing multimodal datasets.

**Baselines** We benchmark ARES against three categories of baselines. First, we include leading proprietary systems such as GPT-4.1 (OpenAI), Gemini-2.5-Pro-Thinking (Comanici et al., 2025), Claude-4-Sonnet (Anthropic), and Doubao-1.5-Thinking-Vision-Pro (Guo et al., 2025), which serve as upper-bound references for multimodal reasoning. Second, we consider representative lightweight open-source MLLMs, including Qwen2.5-VL-3B-Instruct (Bai et al., 2025), FAST-3B (Xiao et al., 2025), and VLAA-Thinker-3B (Chen et al., 2025a), which are suitable for efficient deployment. Third, we evaluate competitive 7B-scale open-source MLLMs, such as Qwen2.5-VL-7B-Instruct, OpenVLThinker-1.2-7B (Deng et al., 2025), MM-Eureka-Qwen-7B (Meng et al., 2025), MMR1-Math-v0 (Leng et al., 2025), ThinkLite-7B-VL (Wang et al., 2025c), VLAA-Thinker-7B, VL-Rethinker-7B (Wang et al., 2025a), and Vision-G1 (Zha et al., 2025). These represent widely adopted open-source alternatives that support research reproducibility. Most of the open-source baselines are fine-tuned from Qwen2.5-VL-3B-Instruct or Qwen2.5-VL-7B-Instruct, ensuring comparability in architecture and training setup. Collectively, this suite provides a comprehensive coverage of both proprietary and open-source models across multiple scales.

## 4.2 MAIN RESULTS

As summarized in Tables 1 and 4, MRLMs consistently outperform chat models across complex reasoning, knowledge-intensive, and general-purpose benchmarks. This confirms that training stages such as cold-start and RLVR effectively enhance self-reflection and reasoning ability. However, we also observe a sharp drop in textual reasoning for many open-source MRLMs likely due to reliance on multimodal verifiable data without cold-start fine-tuning.

In contrast, ARES achieves strong gains at both 3B and 7B scales. On multimodal benchmarks (Table 1), ARES-7B exceeds the best open-source models by +19.0 on MathVision and +11.5 on MMMU-Pro. On textual reasoning (Table 4), it attains 61.7 on AIME25, where most 7B baselines score below 3.3. These results confirm that ARES improves core reasoning ability rather than overfitting to multimodal tasks.

Finally, Table 2 and Figure 3 demonstrate the adaptivity of our training strategy. AdaCS (ARES-CS-7B) modulates response length by task difficulty, while AEPO (ARES-RL-7B) further enhances this effect—extending reasoning for hard tasks (e.g., OlympiadBench, AIME25) and shortening it for easier ones (e.g., GSM8K, MathVista). These results, consistent with our entropy–difficulty analysis (Section 2.2), show that AEPO improves both accuracy and token efficiency by encouraging exploration only when needed.

## 4.3 ABLATION STUDIES

To validate the contributions of the key components within the ARES framework, we conduct a series of ablation studies. We systematically isolate and evaluate the effects of our Hierarchical Reward Design (entropy shaping), and the Dynamic KL Design in our ARES.

Table 1: Performance comparison of various MLLMs on diverse multimodal reasoning benchmarks. Within each model group (3B and 7B), the best results are highlighted in bold, and the second-best are underlined. Scores in *italics* indicate that they are not reported in the original work and are obtained using the VLMEvalKit (Duan et al., 2025) for evaluation. **MathVerse-V**, **DynaMath-W** and **WeMath-S** denotes the vision-only, worst, and strict settings, respectively.

| Model | Multimodal Reasoning Benchmarks | | | | | | | | | | |
|---|---|---|---|---|---|---|---|---|---|---|---|
| | MathVerse-V | MathVision | MathVista | DynaMath-W | WeMath | LogicVista | MMMU | MMMU-Pro | CharXIV | MMStar | Avg. |
| *Close-Source* | | | | | | | | | | | |
| GPT-4.1 | *59.8* | *51.8* | *72.0* | *48.3* | *55.5* | *63.8* | *75.0* | *65.0* | *55.8* | *71.2* | *61.8* |
| Gemini-2.5-Pro-Thinking | *81.2* | *55.3* | *83.8* | *57.1* | *78.0* | *75.2* | *82.0* | *76.5* | *69.3* | *79.7* | *73.8* |
| Claude-4-Sonnet | *66.1* | *54.6* | *70.4* | *46.9* | *63.0* | *64.4* | *74.4* | *60.7* | *58.4* | *66.9* | *62.6* |
| Doubao-1.5-thinking-vision-pro | *80.4* | *68.7* | *85.6* | *60.5* | *78.0* | *71.8* | *77.9* | *67.6* | *63.4* | *78.2* | *73.2* |
| *3B-scale MLLMs* | | | | | | | | | | | |
| Qwen2.5-VL-3B-Instruct | *33.0* | *21.2* | *62.3* | *17.0* | *17.9* | *35.8* | *53.1* | *31.6* | *25.3* | *51.1* | *34.8* |
| FAST-3B | *37.2* | *26.8* | *66.2* | *15.4* | *23.3* | *35.4* | *52.0* | *34.6* | *28.3* | *55.2* | *37.4* |
| VLAA-Thinker-3B | *36.4* | *24.4* | *61.0* | *18.2* | *33.8* | *38.5* | *49.7* | *33.3* | *28.2* | *53.4* | *37.7* |
| **ARES-3B** | **48.2** | **44.2** | **66.8** | **24.6** | **40.8** | **43.5** | **56.8** | **45.2** | **34.4** | **56.7** | **46.1** |
| Δ (Ours–Open 3B SoTA) | +11.0 | +17.4 | +0.6 | +6.4 | +7.0 | +5.0 | +3.7 | +10.6 | +6.1 | +1.5 | +8.4 |
| *7B scale Models* | | | | | | | | | | | |
| Qwen2.5-VL-7B-Instruct | *42.9* | *25.1* | *68.2* | *21.2* | *36.2* | *45.0* | *58.6* | *38.3* | *35.5* | *62.1* | *43.3* |
| OpenVLThinker-1.2-7B | *40.7* | *25.9* | *72.3* | *21.2* | *37.9* | *41.4* | *58.7* | *42.9* | *39.3* | *62.9* | *44.3* |
| MM-Eureka-Qwen-7B | *49.6* | *26.9* | *73.0* | *24.0* | *34.7* | *46.8* | *57.3* | *43.3* | *39.5* | *64.4* | *46.0* |
| MMR1-Math-v0 | *45.1* | *30.2* | *71.0* | *25.2* | *33.2* | *50.8* | *57.1* | *43.2* | *39.3* | *63.8* | *45.9* |
| ThinkLite-7B-VL | *45.3* | *32.9* | *75.1* | *22.0* | *26.5* | *40.7* | *55.5* | *41.3* | *39.3* | *63.4* | *44.2* |
| VLAA-Thinker-7B | *48.2* | *26.4* | *68.0* | *22.4* | *29.2* | *48.5* | *54.6* | *41.6* | *36.1* | *64.5* | *44.0* |
| VL-Rethinker-7B | *49.1* | *32.3* | *74.9* | *27.4* | *27.8* | *44.5* | *56.7* | *41.7* | *39.8* | *62.7* | *45.7* |
| Vision-G1 | *50.0* | *31.3* | **76.1** | *27.2* | *29.0* | *50.2* | *53.4* | *41.2* | *41.0* | *66.0* | *46.5* |
| **ARES-7B** | **56.5** | **51.9** | *74.6* | **39.7** | **47.2** | **54.1** | **67.9** | **54.8** | **47.0** | **65.3** | **55.9** |
| Δ (Ours–7B Open SoTA) | +6.5 | +19.0 | -1.5 | +12.3 | +9.3 | +3.3 | +9.2 | +11.5 | +6.0 | -0.7 | +9.4 |

Figure 3: Training dynamics of accuracy (**left**) and response length (**right**). We observe that ARES achieves higher accuracy compared with its ablations (w/o KL, w/o Entropy) and baseline methods (GRPO, DAPO), while also producing shorter and more stable responses during training. These results indicate that the joint use of KL and entropy shaping contributes to improving both correctness and conciseness.

### 4.3.1 EFFECTIVENESS OF HIERARCHICAL REWARD DESIGN

To verify the impact of our hierarchical entropy-based reward, we conduct an experiment comparing a model trained with only this component against the GRPO baseline. As shown in Table 3, this component alone yields a significant accuracy improvement of **+1.8** points, demonstrating that better depth and valid exploration lead to better performance. Additionally, this adaptive regulation is clearly visible in Figure 3. While the accuracy for this model (labeled "ARES w/o KL") consistently tracks above the GRPO baseline, its response length shows a marked and steady decrease. This shows the model is learning to curtail unnecessary reasoning (i.e., overthinking) on simpler problems. The strong performance gains on difficult benchmarks like DynaMath-W (+3.2 points) further prove it correctly encourages deeper exploration when necessary. This evidence confirms the hierarchical reward's primary role as an adaptive regulator of reasoning depth.

Table 2: **Accuracy and response length comparison across multimodal and textual benchmarks.** We report both accuracy (Acc) and average response length (Len) for five model variants (ARES-CS-Vanilla, ARES-CS-7B, ARES-CS-Vanilla-GRPO, ARES-CS-Vanilla-RL, and ARES-RL-7B) on six benchmarks. Visualization of these results is provided in Figure 8 (accuracy) and Figure 9 (response length) in Appendix.

| Model | Multimodal | | | | | | Textual | | | | | |
|---|---|---|---|---|---|---|---|---|---|---|---|---|
| | OlympiadBench | | MathVerse-V | | MathVista | | AIME25 | | MATH500 | | GSM8K | |
| | Acc | Len | Acc | Len | Acc | Len | Acc | Len | Acc | Len | Acc | Len |
| ARES-CS-Vanilla | 54.2 | 9123.2 | 50.5 | 3432.1 | 72.1 | 1853.3 | 55.2 | 16361.6 | 91.4 | 2750.1 | 92.7 | 358.8 |
| ARES-CS-7B | 53.4 | 10281.5 | 52.8 | 3575.4 | 72.3 | 1712.3 | 55.0 | 17895.0 | 93.4 | 3011.8 | 92.3 | 332.3 |
| ARES-CS-Vanilla-GRPO | 55.8 | 10342.5 | 52.3 | 4256.9 | 71.9 | 2432.5 | 58.2 | 18520.3 | 93.9 | 3185.2 | 93.8 | 395.4 |
| ARES-CS-Vanilla-RL | 56.4 | 11050.4 | 54.7 | 3327.8 | 73.5 | 1573.1 | 60.4 | 20135.4 | 94.1 | 2681.5 | 94.5 | 302.4 |
| ARES-RL-7B | **56.9** | 12893.4 | **56.5** | 3198.3 | **74.6** | 1494.5 | **61.7** | 22618.8 | **95.2** | 2257.7 | **95.7** | 278.9 |

Table 3: **Ablation study of Dynamic KL Loss and Entropy Reward.** Building upon our Cold Start stage. Best results per column are **bold** and second-best are underlined.

| Ablation Setting | MathVerse-V | MathVision | MathVista | DynaMath-W | Avg |
|---|---|---|---|---|---|
| Cold start only | 52.8 | 49.7 | 72.9 | 27.4 | 50.7 |
| Cold Start + GRPO (baseline) | 53.7 | 50.4 | 73.3 | 36.1 | 53.4 |
| Cold Start + DAPO (baseline) | 54.1 | 50.9 | 73.4 | 37.9 | 54.1 |
| Cold Start + Dynamic KL | 55.2 (+2.4) | 51.7 (+2.0) | 73.7 (+0.8) | 38.3 (+10.9) | 54.7 (+4.0) |
| Cold Start + Entropy Shaping | 55.9 (+3.1) | 51.3 (+1.6) | 74.4 (+1.5) | 39.3 (+11.9) | 55.2 (+4.5) |
| ARES-7B | **56.5** (+3.7) | **51.9** (+2.2) | **74.6** (+1.7) | **39.7** (+12.3) | **55.7** (+5.0) |

### 4.3.2 EFFECTIVENESS OF DYNAMIC KL DESIGN

To verify that the dynamic KL mechanism allocates the exploration budget efficiently, we analyze its isolated contribution. The results in Table 3 show that adding only the dynamic KL component improves average accuracy by **+1.3** points over the GRPO baseline. Its role as an efficient budget allocator is best illustrated in Figure 3. The full ARES model's accuracy curve ultimately converges to a higher final accuracy than all other variants. Concurrently, its response length curve shows the most pronounced reduction, indicating that the token-wise budget allocation pushes the model to find more efficient reasoning paths. This outcome validates that the dynamic KL design successfully allocates the thinking budget, guiding the policy toward greater reasoning efficiency and higher accuracy.

Finally, combining the hierarchical reward with the dynamic KL mechanism results in the full ARES framework. As shown in Table 3, this complete model achieves the highest average accuracy of all configurations (55.7). This result is corroborated by Figure 3, where we see the full ARES model not only converges to the highest final accuracy but also achieves the most significant reduction in response length. This demonstrates the synergistic effect of our two components: by simultaneously regulating reasoning depth and efficiently allocating the exploration budget, ARES achieves both stronger accuracy and more efficient training.

## 5 RELATED WORK

### 5.1 MULTIMODAL LARGE REASONING MODELS

Reasoning is widely recognized as a foundational element of intelligent behavior (de Winter et al., 2024; Bi et al., 2025). To meet the demands of realistic multimodal environments, Multimodal Large Reasoning Models (MLRMs) fuse information from heterogeneous modalities to support deeper thinking. Extending chain-of-thought (CoT) fine-tuning to instill stepwise reasoning, works such as LLaVA-CoT (Xu et al., 2025) and LlamaV-o1 (Thawakar et al., 2025) transpose this paradigm to the multimodal setting. Following the success of DeepSeek-R1 (DeepSeek-AI et al., 2025), a growing body of research leverages reinforcement learning (RL) to extend long-horizon reasoning in MLRMs (Peng et al., 2025; Shen et al., 2025a). Against this backdrop, our goal is to build an adaptive MLRM that can decide how much reasoning effort to allocate at inference time.

| | | Textual Reasoning Benchmarks | | | | | |
|---|---|---|---|---|---|---|---|
| Model | AIME24 | AIME25 | MATH500 | MMLU Pro | BBEH | GPQA | Avg. |
| *Closed-Source* | | | | | | | |
| GPT-4.1 | 46.7 | 23.3 | 89.6 | 80.2 | 32.7 | 67.7 | 56.7 |
| Gemini-2.5-Pro-Thinking | 66.7 | 88.0 | 85.8 | 84.3 | 17.7 | 82.3 | 70.8 |
| Claude-4-Sonnet | 46.7 | 33.3 | 92.0 | 83.2 | 35.0 | 68.7 | 59.8 |
| Doubao-1.5-Thinking-Vision-Pro | 80.0 | 53.3 | 96.8 | 83.5 | 39.7 | 71.2 | 70.8 |
| *3B-scale MLLMs* | | | | | | | |
| Qwen2.5-VL-3B-Instruct | 0.0 | 0.0 | 55.8 | **41.5** | 7.0 | **29.3** | 22.3 |
| FAST-3B | 6.7 | 3.3 | 55.4 | 34.6 | **8.6** | 25.3 | 22.3 |
| VLAA-Thinker-3B | 3.3 | 0.0 | 61.4 | 33.3 | 8.1 | 27.8 | 22.3 |
| **ARES-3B** | **41.5** | **34.8** | **86.2** | 55.8 | 15.0 | 41.4 | **45.8** |
| Δ (Ours–Open 3B SoTA) | +34.8 | +31.5 | +24.8 | +14.3 | +6.4 | +12.1 | +20.7 |
| *7B-scale MLLMs* | | | | | | | |
| Qwen2.5-VL-7B-Instruct | 6.7 | 0.0 | 66.0 | 50.5 | 10.8 | 36.9 | 28.5 |
| OpenVLThinker-1.2-7B | 0.0 | 0.0 | 63.2 | 49.3 | 13.6 | 31.3 | 26.2 |
| MM-Eureka-Qwen-7B | 13.3 | 0.0 | 65.4 | **53.4** | 12.2 | **41.4** | 31.0 |
| MMR1-Math-v0 | 3.3 | 3.3 | 66.6 | 51.8 | 11.4 | 32.3 | 28.1 |
| ThinkLite-7B-VL | 3.3 | 3.3 | 67.6 | 51.9 | 11.3 | 35.9 | 28.9 |
| VLAA-Thinker-7B | 6.7 | 3.3 | 68.6 | 51.2 | 11.7 | 28.8 | 28.4 |
| VL-Rethinker-7B | 3.3 | 3.3 | 64.4 | 52.0 | 12.9 | 31.8 | 28.0 |
| Vision-G1 | 3.3 | 0.0 | 69.0 | 53.0 | **13.9** | 31.3 | 28.4 |
| **ARES-7B** | **65.0** | **61.7** | **95.2** | **67.0** | **18.9** | **49.5** | **59.6** |
| Δ (Ours–7B Open SoTA) | +51.7 | +58.4 | +26.2 | +13.6 | +5.0 | +8.1 | +27.2 |

Table 4: **Performance on textual reasoning benchmarks.** AIME24 and AIME25 results are averaged over eight independent inference runs to reduce score variance. Results on AIME24/25, MATH500, MMLU Pro, BBEH, and GPQA. ARES-3B and ARES-7B substantially outperform all open-source baselines at their respective scales, achieving large average gains (Δ rows) and narrowing the gap to leading proprietary systems.

## 5.2 ADAPTIVE REASONING

Current large reasoning models such as OpenAI o1 (OpenAI et al., 2024) and DeepSeek R1 (DeepSeek-AI et al., 2025) show notable reasoning capabilities while often generate excessively lengthy reasoning for trivial questions while providing insufficient exploration for more challenging ones (Zhu & Li, 2025; Alomrani et al., 2025). To address this limitation, numerous studies have sought to establish adaptive reasoning mechanisms. One line of research develops training-free approaches based on prompt engineering (Jiang et al., 2025; Yang et al., 2025a). Complementary to these approaches, training-based methods aim to instill more strategic adaptive reasoning. Some studies curate variable-length datasets containing both concise and elaborate reasoning chains for supervised fine-tuning (SFT) (Yu et al., 2025a; Ma et al., 2025), while others leverage reinforcement learning (RL) to encourage dynamic control over reasoning depth by integrate length penalties (Arora & Zanette, 2025; Tu et al., 2025) or employ difficulty-awareness mechanism (Chen et al., 2025c; Xiang et al., 2025). However, most work still focuses on text-only reasoning, and these approaches often over-encourage exploration on hard problems. In this work, we take an initial step toward multimodal adaptive reasoning framework, and treat entropy as a localized signal to encourage concise solutions for easy questions and sustained exploration on hard ones.

## 6 CONCLUSION

This paper introduces ARES, a framework designed to cultivate adaptive reasoning in MLRMs, addressing their tendency to overthink simple problems while under-exploring complex ones. ARES uses a two-stage training curriculum: an Adaptive Cold-Start phase instills difficulty awareness, followed by our novel Adaptive-Entropy Policy Optimization (AEPO). AEPO leverages high window-entropy to determine when to explore and a hierarchical reward to control how much reasoning depth is applied. Experiments confirm that ARES achieves superior performance and significantly improves reasoning efficiency, validating our adaptive, entropy-guided approach.

REPRODUCIBILITY STATEMENT

To facilitate reproducibility, we provide the full implementation of our ARES framework in an anonymous code repository. Our models are built upon the publicly available Qwen2.5-VL-3B and Qwen2.5-VL-7B-Instruct. The datasets used for all training and evaluation stages, including the Adaptive Cold-Start and AEPO phases, are curated from public sources. Furthermore, the Appendix provides a comprehensive breakdown of our data curation process, experimental setup, and key hyperparameters to ensure our results can be replicated.

ETHICS STATEMENT

We are not aware of any major ethical concerns arising from our work. Our study is conducted entirely within the academic domain of multimodal reasoning, using only publicly available models and datasets for training and evaluation. No human subjects were involved, and our research does not introduce sensitive or potentially harmful capabilities. The objective of ARES is to advance research on adaptive and efficient reasoning for the open-source community, and we do not foresee any direct societal risks from this work.

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

APPENDIX

APPENDIX CONTENTS

# A  LLM USAGE STATEMENT

In this paper, we employed a Large Language Model (LLM) in an assistive capacity for this manuscript. Its use was strictly limited for grammatical accuracy and refining sentence structure to improve clarity and readability. All intellectual contributions, including the formulation of ideas, data analysis, and the composition of the manuscript, are entirely the work of the authors.

## B EXPERIMENTAL SETTINGS

### B.1 BENCHMARKS

To comprehensively evaluate the capabilities of our model, we selected a diverse suite of benchmarks designed to test a wide range of reasoning skills. In the multimodal domain, we assess multimodal mathematical reasoning using MathVerse (Zhang et al., 2024a), MathVision (Wang et al., 2024a), MathVista (Lu et al., 2023), DynaMath (Zou et al., 2025), and WeMath (Qiao et al., 2024). To evaluate broader logical and general-purpose multimodal capabilities, we incorporated LogicVista (Xiao et al., 2024), MMMU (Yue et al., 2024a), MMMU-Pro (Yue et al., 2024b), CharXIV (Wang et al., 2024c), and MMStar Chen et al. (2024). For text-based reasoning, we measured the model's ability to solve challenging mathematical problems with AIME24/25 (Li et al., 2024a) and MATH-500 (Hendrycks et al., 2021), while its general question-answering and reasoning proficiency were tested on GPQA (Rein et al., 2024), MMLU Pro (Wang et al., 2024b), and BBEH (Kazemi et al., 2025). For all evaluations, we report pass@1 accuracy, with the exception of the AIME24/25 benchmark, for which performance is measured using average@16 accuracy.

### B.2 IMPLEMENTATION DETAILS

Our models, based on Qwen2.5-VL-7B-Instruct and Qwen2.5-VL-3B-Instruct, are trained using a two-stage strategy. The first stage is a cold-start supervised fine-tuning (SFT), where we independently fine-tune the large language model (LLM) module for two epochs. This stage uses a batch size of 256, a sequence length of 32k, and a learning rate of $2 \times 10^{-5}$. Following SFT, we further optimize the model using Adaptive Entropy Policy Optimization (AEPO). In each AEPO iteration, we sample 512 prompts from the training set and generate eight rollouts per prompt, yielding 4,096 trajectories. Rollouts are generated with a temperature of 1.0 and a top-p of 0.99, and the maximum sequence length is set to 20k tokens (a 4k token prompt and a 16k token response). For policy updates, we use a global batch size of 128 and the AdamW optimizer with a learning rate of $1 \times 10^{-6}$ and a weight decay of $1 \times 10^{-2}$, without a learning rate warmup. To ensure training stability, we apply online filtering to discard samples whose mean overall reward falls outside the predefined range of [0.01, 0.99]. This process removes outliers that could distort advantage estimation and impede policy updates.

**Compute Efficiency.** Stable-MM-R1 requires $N = 16$ rollouts per query during the initial PAQM phase. While this incurs a one-time preprocessing cost (approx. $1.5\times$ standard epoch time), the resulting training stability allows for faster convergence. As shown in Figure **??**, our method reaches peak performance in fewer steps than baselines, effectively amortizing the initial rollout cost.

Table 5: Textual and multimodal reasoning datasets source of ARES cold-start data.

| Source | Multimodal | | | | Text-only | | |
|---|---|---|---|---|---|---|---|
| | Samples | Source | Samples | Source | Samples | Source | Samples |
| FigureQA (Kahou et al., 2017) | 100K | Super-CLEVR (Li et al., 2023) | 30K | Big-Math-RL (Albalak et al., 2025) | 251K | GAIR_LIMO (Ye et al., 2025) | 0.8K |
| MAVIS (Zhang et al., 2024b) | 218K | TabMWP (Lu et al., 2022) | 38K | Big-Math-RL-U (Albalak et al., 2025) | 35K | s1K-1.1 (Muennighoff et al., 2025) | 1K |
| GeoQA (Chen et al., 2021) | 5K | UniGeo (Chen et al., 2022) | 16K | OpenThoughts (Guha et al., 2025) | 114K | OpenMathR (Moshkov et al., 2025) | 3,200K |
| Geometry3K (Lu et al., 2021a) | 2.1K | MultiMath (Peng et al., 2024) | 300K | DeepMath (He et al., 2025) | 103K | OrcaMath (Mitra et al., 2024) | 200K |
| IconQA (Lu et al., 2021b) | 107K | Grammer (Chen et al., 2025b) | 30K | OpenR1-220k (Face, 2025) | 220K | NuminaMath-CoT (Li et al., 2024b) | 859K |

## C TOKEN-LEVEL ENTROPY MEASUREMENT

To quantify the model's local uncertainty during generation, we compute the *token-level entropy* at step $t$ as

$$H_t = -\sum_{j=1}^{V} p_{t,j} \log p_{t,j}, \qquad p_t = \pi_\theta(\cdot \mid q, o_{<t}), \tag{3}$$

where $\pi_\theta$ denotes the policy under parameters $\theta$, $q$ is the input query, and $o_{<t}$ is the prefix of generated tokens up to position $t-1$. The probability vector $p_t \in \mathbb{R}^V$ is obtained from the normalized logits $z_t$ by a temperature-scaled softmax transformation:

$$p_t = \text{Softmax}\left(\tfrac{z_t}{T}\right),$$

with vocabulary size $V$ and decoding temperature $T$ controlling the sharpness of the distribution.

**Role in our setting.** Unlike sequence-level measures (e.g., log-likelihood of the whole output), $H_t$ captures the fine-grained uncertainty at each generation step. During reinforcement learning, we treat $H_t$ as an intrinsic signal of "reasoning hesitation": a higher entropy indicates that the model considers multiple plausible continuations, whereas a lower entropy suggests that the model is confident about the next token.

Note that $H_t$ is tied to the distribution $p_t$, not to the realized token $o_t$ drawn from $p_t$. Hence two identical tokens generated at different timesteps may correspond to different entropies, depending on the local distributional context. In this sense, token entropy should be interpreted as a property of the model's decision process rather than of the discrete outcome itself.

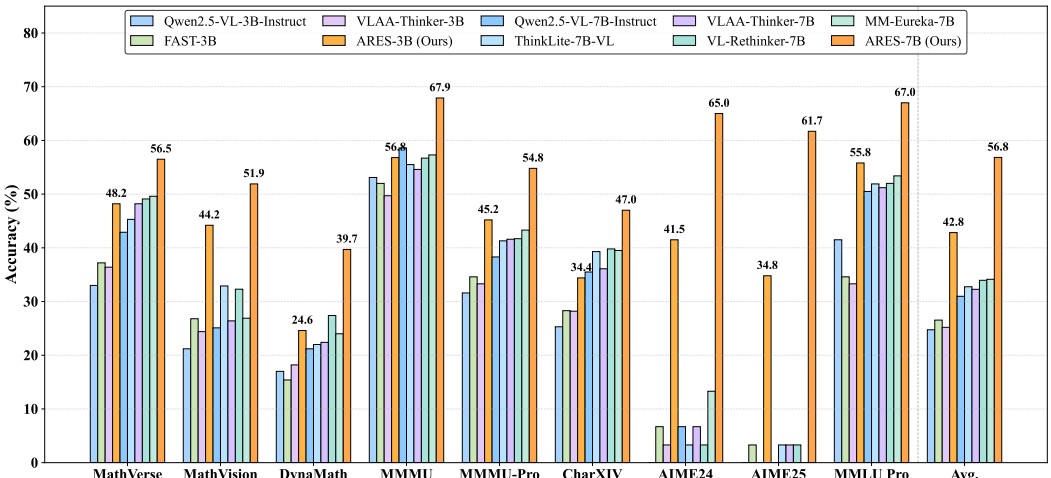

Figure 4: Accuracy comparison across selected open-source reasoning models on nine multimodal and textual benchmarks. Each group represents 3B-scale and 7B-scale models evaluated under the same benchmarks. The rightmost column ("Avg.") reports the average accuracy over all selected benchmarks, showing the overall advantage of the proposed adaptive reasoning framework. Our **ARES-7B** achieves superior performance.

# D RLVR ALGORITHMS

In this subsection, we provide additional details of the baseline RLVR algorithms used in our experiments, namely *Group Relative Policy Optimization (GRPO)* and *Dynamic sAmpling Policy Optimization (DAPO)*. Both methods extend traditional policy optimization frameworks to improve the stability and effectiveness of reinforcement learning for reasoning tasks, yet they differ in how they handle variance reduction, KL regularization, and sampling dynamics.

## D.1 GROUP RELATIVE POLICY OPTIMIZATION (GRPO).

GRPO (DeepSeek-AI et al., 2025) extends conventional policy optimization by introducing group-wise normalization. Specifically, training samples are divided into $K$ groups $\{\mathcal{G}_1, \ldots, \mathcal{G}_K\}$ based on input prompts or task-specific criteria. For each group $\mathcal{G}_i$, both a policy $\pi_\theta$ and a frozen reference policy $\pi_{\theta_{\mathrm{ref}}}$ are maintained. The GRPO objective can be written as:

$$\mathbb{E}_{x \sim \mathcal{G}_i,\, y \sim \pi_\theta(y|x)} \left[ \min \left( \frac{\pi_\theta(y|x)}{\pi_{\theta_{\mathrm{ref}}}(y|x)} \, \hat{A}(x,y), \ \mathrm{clip}\left( \frac{\pi_\theta(y|x)}{\pi_{\theta_{\mathrm{ref}}}(y|x)}, \, 1 - \epsilon, \, 1 + \epsilon \right) \hat{A}(x,y) \right) \right],$$

where $\epsilon$ controls the size of the trust region. The group-relative advantage $\hat{A}(x,y)$ is defined as:

$$\hat{A}(x, y_i) = \frac{r(x, y_i) - \mathrm{mean}(\{r(x, y_1), \ldots, r(x, y_G)\})}{\mathrm{std}(\{r(x, y_1), \ldots, r(x, y_G)\}) + \epsilon}.$$

This relative advantage stabilizes training by normalizing each sample's reward against the variance of its group. By leveraging intra-group normalization, GRPO encourages responses that are better than their peers, while maintaining diversity across groups.

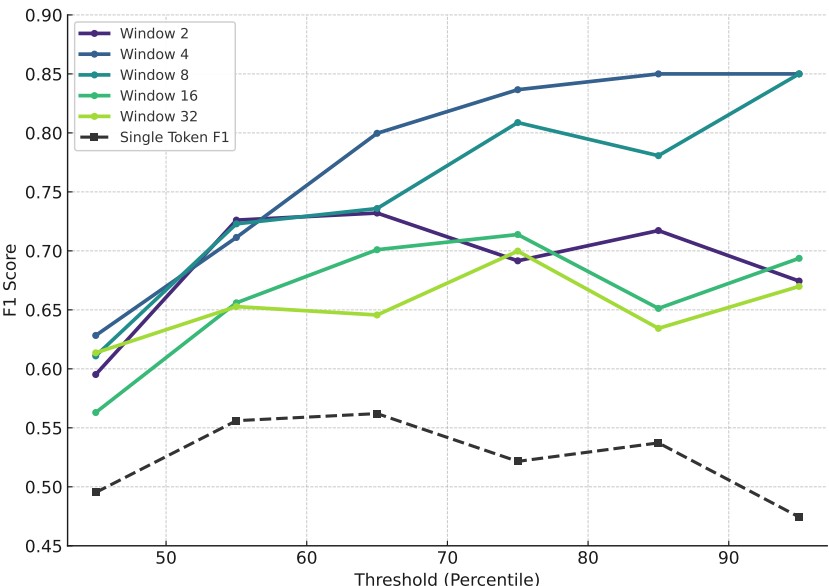

Figure 5: F1 Score vs. threshold percentile across different window sizes. Window-based entropy aggregation consistently outperforms single-token entropy, especially at higher thresholds.

## D.2 DYNAMIC SAMPLING POLICY OPTIMIZATION (DAPO).

Building on GRPO (DeepSeek-AI et al., 2025), DAPO (Yu et al., 2025b) removes the explicit KL penalty and instead incorporates several mechanisms for improved sample efficiency. First, it introduces a *clip-higher* strategy to reduce over-penalization of promising trajectories. Second, it adopts *dynamic sampling*, where response sampling probabilities are adaptively adjusted based on entropy signals. Third, DAPO employs token-level policy gradient updates with overlong reward shaping. The DAPO objective is formulated as:

$$
\mathcal{J}_{\text{DAPO}}(\theta) = \mathbb{E}_{(q,a)\sim\mathcal{D},\,\{o^i\}_{i=1}^G\sim\pi_{\theta_{\text{old}}}(\cdot|q)} \left[ \frac{1}{\sum_{i=1}^G |o^i|} \sum_{i=1}^G \sum_{t=1}^{|o^i|} \min\left(r_t^i(\theta),\, \text{clip}(r_t^i(\theta), 1-\epsilon_{\text{low}}, 1+\epsilon_{\text{high}})\,\hat{A}_t^i\right) \right],
$$

where $r_t^i(\theta)$ is defined as in GRPO, and $\hat{A}_t^i$ is the token-level relative advantage. Unlike GRPO, which focuses on group-level normalization, DAPO shifts optimization to the token level, enabling finer-grained control and adaptive shaping of the learning signal. This makes DAPO a strong baseline for RLVR without requiring a value network.

Both GRPO and DAPO contribute to reducing variance and stabilizing policy optimization in RLVR. GRPO leverages group-relative normalization, while DAPO refines the approach with token-level updates and dynamic sampling. In our work, we adopt DAPO as the primary baseline due to its strong empirical performance and its widespread use in prior RLVR literature.

## E WINDOW-ENTROPY AGGREGATION

While token-level entropy $H_t$ provides a fine-grained measure of uncertainty, single-token fluctuations are often dominated by lexical ambiguity or local randomness. To capture more coherent reasoning segments, we aggregate entropy over a sliding window of length $w$:

$$
\bar{H}_{t:w} \;=\; \frac{1}{w} \sum_{\tau=t}^{t+w-1} H_\tau. \tag{4}
$$

This windowed statistic smooths out spurious token-level spikes and highlights regions where the model persistently maintains high uncertainty.

A window is flagged as *high-entropy* if its average entropy exceeds a dynamic threshold $\theta$, which is updated online to match the distributional scale of the current batch, which is shown in Section 2.1. Intuitively, these windows correspond to stretches of the output where the model is genuinely uncertain about its reasoning trajectory, as opposed to momentary ambiguity on function words or symbols.

The aggregated entropy $\bar{H}_{t:w}$ therefore serves two purposes: (i) it localizes "hard thinking" phases within the output, providing a natural signal for allocating additional exploration, and (ii) it stabilizes training by reducing sensitivity to token-level noise. In our AEPO algorithm, only tokens falling inside validated high-entropy windows are granted relaxed KL budgets, thereby ensuring that extra computation is spent precisely where reasoning effort is most needed.

## F  KL Penalty Inflates GRPO Advantage Variance compared to KL Loss

For a fixed prompt $x$, GRPO samples a group of $N$ responses $\{y_i\}_{i=1}^N$ i.i.d. from $\pi_\theta(\cdot \mid x)$. Let the *task+entropy shaped return* be

$$S_i \triangleq r_{\text{acc}}(x, y_i) - \lambda_d \frac{C_{\text{ent}}(x, y_i)}{Z_{\text{ent}}}, \tag{5}$$

and the sample-averaged per-token reference KL estimator be

$$K_i \triangleq \frac{1}{L} \sum_{t=1}^{L} \text{kld}_{t,i}. \tag{6}$$

We compare two GRPO implementations:

1. **KL penalty** (merge KL into the return):

$$R_i' = S_i - \kappa K_i, \qquad A_i' = R_i' - \bar{R}', \quad \bar{R}' = \tfrac{1}{N} \sum_{j=1}^{N} R_j'.$$

2. **KL loss (actor-only)**:

$$R_i = S_i, \qquad A_i = R_i - \bar{R}, \quad \bar{R} = \tfrac{1}{N} \sum_{j=1}^{N} R_j,$$

and add the separate actor regularizer $\kappa \cdot \frac{1}{L} \sum_t \beta_t \text{kld}_{t,i}$ which does *not* enter $A_i$.

Assume $\{(S_i, K_i)\}_{i=1}^N$ are i.i.d. with finite second moments, and denote

$$\sigma_S^2 = \text{Var}(S), \qquad \sigma_K^2 = \text{Var}(K), \qquad \text{Cov}(S, K) = \rho\,\sigma_S \sigma_K.$$

**Lemma 1** (Group-baseline advantage variance). *For i.i.d. $R_1, \ldots, R_N$ with finite variance, the GRPO group-baseline advantage $A_i = R_i - \bar{R}$ satisfies*

$$\text{Var}(A_i) = \left(1 - \tfrac{1}{N}\right) \text{Var}(R). \tag{7}$$

*Proof.* Write $A_i = (1 - \frac{1}{N})R_i - \frac{1}{N} \sum_{j \neq i} R_j$ and use i.i.d. plus independence across indices to expand $\text{Var}(A_i)$; cross-terms cancel and equation 7 follows. □

**Proposition 1** (KL penalty inflates GRPO advantage variance). *Under the assumptions above,*

$$\text{Var}(A_i') - \text{Var}(A_i) = \left(1 - \tfrac{1}{N}\right) \left(\kappa^2 \sigma_K^2 - 2\kappa \,\text{Cov}(S, K)\right). \tag{8}$$

*In particular, if $\text{Cov}(S, K) \approx 0$ (empirically common), then $\text{Var}(A_i') - \text{Var}(A_i) \approx \left(1 - \frac{1}{N}\right)\kappa^2 \sigma_K^2 > 0$ for any $\kappa \neq 0$. More generally, unless $\kappa \in \left(0, 2\,\text{Cov}(S, K)/\sigma_K^2\right)$ (a narrow interval when $\text{Cov}(S, K)$ is small), one has $\text{Var}(A_i') > \text{Var}(A_i)$.*

*Proof.* By Lemma 1, it suffices to compare $\text{Var}(R_i')$ to $\text{Var}(R_i)$. We have $\text{Var}(R_i) = \text{Var}(S) = \sigma_S^2$ and

$$\text{Var}(R_i') = \text{Var}(S_i - \kappa K_i) = \sigma_S^2 + \kappa^2 \sigma_K^2 - 2\kappa \,\text{Cov}(S, K).$$

Subtracting and multiplying by $(1 - \frac{1}{N})$ yields equation 8. □

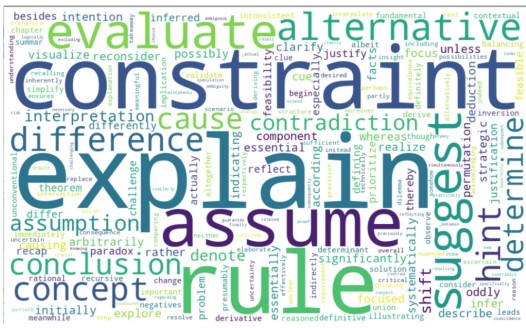

Figure 6: A word cloud visualization of semantically filtered high-entropy tokens, where font size reflects relative frequency. These tokens (e.g., *explain*, *assume*, *constraint*, *conclude*) correspond to reasoning triggers that mark the onset of logical transitions, highlighting the interpretable basis of our entropy-based reward.

**Consequences for gradient variance.** Let the score-function term be $G_i = \sum_{t=1}^{L} \nabla_\theta \log \pi_\theta(y_{i,t} \mid s_{i,t})$ and suppose, as is standard in variance analyses, that $A$ and $G$ are weakly correlated.[1] Then

$$\mathrm{Var}(A_i' G_i) \approx \mathrm{Var}(A_i') \mathbb{E}\|G_i\|^2 > \mathrm{Var}(A_i) \mathbb{E}\|G_i\|^2 \approx \mathrm{Var}(A_i G_i),$$

whenever $\mathrm{Var}(A_i') > \mathrm{Var}(A_i)$ by Proposition 1. In the KL-loss implementation, the additional KL gradient $\kappa \sum_t \beta_t \nabla_\theta \mathrm{kld}_{t,i}$ is *not* multiplied by the group advantage and therefore does not inherit its variance amplification; it also targets the trust-region deviation at a per-token granularity, improving credit assignment.

Merging per-token KL into the return injects a high-frequency, task-agnostic signal ($K_i$) into the groupwise competition, thereby amplifying the variance of the advantage and the policy-gradient estimator. Using KL as an actor-only regularizer decouples the trust-region control from the task/entropy signals, keeping the advantage low-variance and allowing a dedicated controller to track the KL budget.

## G  THE ALGORITHM WORKFLOW OF AEPO

In this section, we provide a detailed flowchart of our AEPO algorithm in the following diagram 1.

## H  ON-POLICY DIFFICULTY AND BUCKETS

Instance difficulty is estimated online using group rollouts. With $K=8$ samples per prompt,

$$\mathrm{pass@8}(x) = \frac{1}{8} \sum_{k=1}^{8} \mathbf{1}\{\mathrm{correct}(y^{(k)}, x)\}, \tag{9}$$

and we assign buckets

$$d(x) = \begin{cases} \texttt{easy}, & \mathrm{pass@8}(x) \geq 6, \\ \texttt{medium}, & 3 \leq \mathrm{pass@8}(x) < 6, \\ \texttt{hard}, & \mathrm{pass@8}(x) \leq 2. \end{cases} \tag{10}$$

Buckets are recomputed each iteration and will parameterize per-bucket budgets/targets in our method.

## I  HIGH-ENTROPY TOKENS

In our analysis, we define the number of *high-entropy tokens* in a trajectory by combining an entropy-based thresholding procedure with a semantics-aware filtering step.

---

[1]This assumption is used for *comparative* purposes; the conclusion also holds under more general covariance decompositions.

---

**Algorithm 1:** Adaptive-Entropy Policy Optimization (AEPO)

---

**Input** : Reference model $\pi_{\text{ref}}$; initial policy $\pi_\theta$; prompts $\mathcal{D}$;
group size $G$; window size $w$; rollout parameters (temperature, top-$p$);
entropy quantile $q = 0.95$; relaxation factor $\rho < 1$;
difficulty buckets $\{\texttt{easy}, \texttt{medium}, \texttt{hard}\}$ with KL targets $\{\delta_d\}$,
entropy weights $\{\lambda_d\}$, base KL weights $\{\beta_d\}$, and shaping functions $S_d(\cdot)$;
GRPO clipping parameter $\epsilon$; controller step size $\alpha_\kappa$.
**Output:** Updated policy $\pi_\theta$

---

1 **for** *each training iteration* **do**
2     Sample a mini-batch of prompts $\{x_b\}_{b=1}^B \sim \mathcal{D}$
3     **foreach** *prompt $x$* **do**
       `// (1) Rollout generation and entropy computation`
4        Generate $G$ rollouts $\{o_i\}_{i=1}^G \sim \pi_\theta(\cdot \mid x)$
5        Compute token entropies $H_{i,t}$ and sliding-window means $\bar{H}_{i,t:w}$
       `// (2) Dynamic thresholding`
6        $\theta_i \leftarrow \text{Quantile}_q(\{H_{i,t}\}_t)$ for each $i$;
7        $\theta \leftarrow \frac{1}{G}\sum_{i=1}^G \theta_i \; m_{i,t} \leftarrow \mathbb{I}\{\bar{H}_{i,t:w} \geq \theta\}; \quad N_{\text{HE}}^{(i)} \leftarrow \sum_t m_{i,t}$
       `// (3) Difficulty estimation`
8        Compute pass@$G$ accuracy $p_x$; assign difficulty bucket $d(x)$ based on $p_x$
       `// (4) Sequence rewards with entropy shaping`
9        $R_i \leftarrow r_{\text{acc}}(x, o_i) + S_{d(x)}(N_{\text{HE}}^{(i)}, \text{acc}_i) - \lambda_{d(x)} \frac{C_{\text{ent}}^{(i)}}{Z_i}$
       `// (5) Group-centered advantages with entropy bonus`
10        $\widehat{R}_i \leftarrow R_i - \frac{1}{G}\sum_{j=1}^G R_j$
11        **for** $t = 1$ **to** $L_i$ **do**
12           $A_{i,t}^{\text{grp}} \leftarrow \widehat{R}_i / L_i$
13           $\psi_{i,t} \leftarrow \lambda_{d(x)} \, \phi([\bar{H}_{i,t:w} - \theta]_+) \, m_{i,t} - b_i$
14           $\tilde{A}_{i,t} \leftarrow A_{i,t}^{\text{grp}} + \psi_{i,t}$
15        **end**
       `// (6) KL regularization with window-adaptive weights`
16        $\text{kld}_{i,t} \leftarrow D_{\text{KL}}(\pi_\theta(\cdot \mid s_{i,t}) \| \pi_{\text{ref}}(\cdot \mid s_{i,t}))$
17        $\beta_{i,t} \leftarrow \beta_{d(x)} \cdot \rho^{m_{i,t}}$
       `// (7) Actor update with clipped GRPO objective`
18        $r_{i,t}(\theta) \leftarrow \pi_\theta(o_{i,t} \mid s_{i,t}) / \pi_{\theta_{\text{old}}}(o_{i,t} \mid s_{i,t})$
19        $\mathcal{L}_{\text{actor}} \leftarrow -\frac{1}{G}\sum_i \frac{1}{L_i}\sum_t \min\{r_{i,t}(\theta)\tilde{A}_{i,t}, \text{clip}(r_{i,t}(\theta), 1-\epsilon, 1+\epsilon)\tilde{A}_{i,t}\}$
20        $\mathcal{L}_{\text{KL}} \leftarrow \kappa_{d(x)} \cdot \frac{1}{G}\sum_i \frac{1}{L_i}\sum_t \beta_{i,t} \, \text{kld}_{i,t}$
21        $\mathcal{L} \leftarrow \mathcal{L}_{\text{actor}} + \mathcal{L}_{\text{KL}}$
22        Update $\theta \leftarrow \theta - \eta \nabla_\theta \mathcal{L}$
       `// (8) KL controller update (non-window tokens only)`
23        $\text{KL}_{\text{ctrl}} \leftarrow \frac{1}{G}\sum_i \frac{1}{L_i}\sum_t \mathbb{I}\{m_{i,t} = 0\} \, \text{kld}_{i,t}$
24        $\kappa_{d(x)} \leftarrow \text{clip}\Big(\kappa_{d(x)}\big(1 + \alpha_\kappa(\text{KL}_{\text{ctrl}}/\delta_{d(x)} - 1)\big), \kappa_{\min}, \kappa_{\max}\Big)$
25     **end**
26 **end**

---

**Semantic filtering.** Rather than counting all tokens that exceed a certain entropy threshold, we first restrict attention to a fixed vocabulary of *semantically meaningful reasoning triggers*. This vocabulary is constructed using a filtering prompt applied to GPT, which classifies each candidate token as either "reasoning-relevant" (True) or not (False). The retained tokens include transition markers (e.g., *but*, *however*), reasoning initiators (e.g., *therefore*, *thus*), and other structural markers that signal the onset of a reasoning step. Tokens with little semantic content (e.g., *A*, *B*, digits, or domain-specific placeholders) are excluded. The entire resulting vocabulary can be visualized in Figure 6, where the word cloud highlights the most frequent reasoning triggers.

**Entropy thresholding.** Given token-level entropy values $\{H_t\}_{t=1}^T$, we compute the 95th percentile of entropies within the current batch, denoted $\theta_{0.95}$. A token $o_t$ is labeled as *high-entropy* if

$$H_t \geq \theta_{0.95} \quad \text{and} \quad o_t \in \mathcal{V}_{\text{sem}},$$

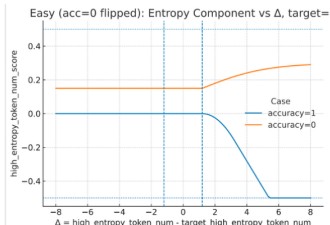 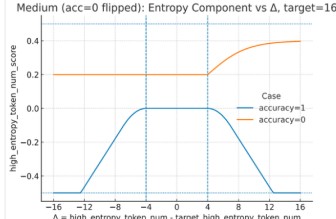 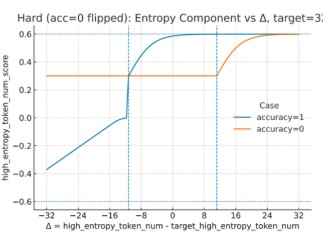

Figure 7: **Effect of entropy shaping and KL regularization on accuracy reward.** We plot the moving-average accuracy reward over training steps under different ablation settings. Baseline methods like GRPO and DAPO either lack stable improvement or plateau early. In contrast, our ARES variants consistently achieve higher accuracy rewards throughout training. Notably, combining both KL regularization and entropy shaping yields the most stable and significant gains, demonstrating the necessity of the two components working together.

where $\mathcal{V}_{\text{sem}}$ is the fixed set of semantically filtered tokens.

The number of high-entropy tokens in a trajectory $y = (o_1, \ldots, o_T)$ is thus defined as

$$N_{\text{HE}}(y) = \sum_{t=1}^{T} \mathbf{1}[H_t \geq \theta_{0.95} \ \wedge \ o_t \in \mathcal{V}_{\text{sem}}].$$

This definition ensures that our entropy-based reward focuses only on tokens that both exhibit unusually high uncertainty and carry clear semantic signals of reasoning transitions. As a result, the entropy reward mechanism becomes more interpretable and more closely aligned with reasoning-relevant exploratory behavior.

## J  VISUAL ANALYSIS OF ENTROPY REWARD DESIGN

To better understand the effect of our entropy reward shaping, we provide a visual analysis in Figure 7. Each plot illustrates the entropy reward component as a function of the deviation $\Delta = N_{\text{HE}} - N_{\text{HE}}^{\text{target}}$, where $N_{\text{HE}}$ is the number of detected high-entropy tokens and $N_{\text{HE}}^{\text{target}}$ is the difficulty-dependent target.

**Easy tasks.** For easy problems, excessive high-entropy activity indicates unnecessary overthinking. As shown in the left panel, the entropy reward heavily penalizes positive deviations ($\Delta > 0$), while providing only mild encouragement when errors occur (acc = 0). This ensures that correct responses remain short and efficient, while still allowing a small degree of exploratory thinking if the model initially fails.

**Medium tasks.** For medium-difficulty problems, the design is symmetric around the target. Moderate deviations from the target are tolerated, but excessive deviations in either direction are penalized. Importantly, when the model answers incorrectly, the entropy reward encourages it to increase reasoning length (i.e., generate more high-entropy tokens). This balances efficiency with robustness, preventing both under- and over-thinking.

**Hard tasks.** For hard problems, high entropy is consistently rewarded regardless of accuracy. As shown in the right panel, both correct and incorrect responses receive positive shaping when $\Delta > 0$. This reflects the intuition that difficult problems require longer and more exploratory reasoning chains, and the model should be encouraged to sustain high-entropy reasoning segments.

Overall, the entropy reward curves realize a difficulty-aware shaping strategy: they suppress unnecessary verbosity on easy problems, regulate reasoning depth for medium ones, and strongly encourage exploratory thinking on hard problems. This design provides a smooth and interpretable mechanism for adaptive reasoning across the full difficulty spectrum.

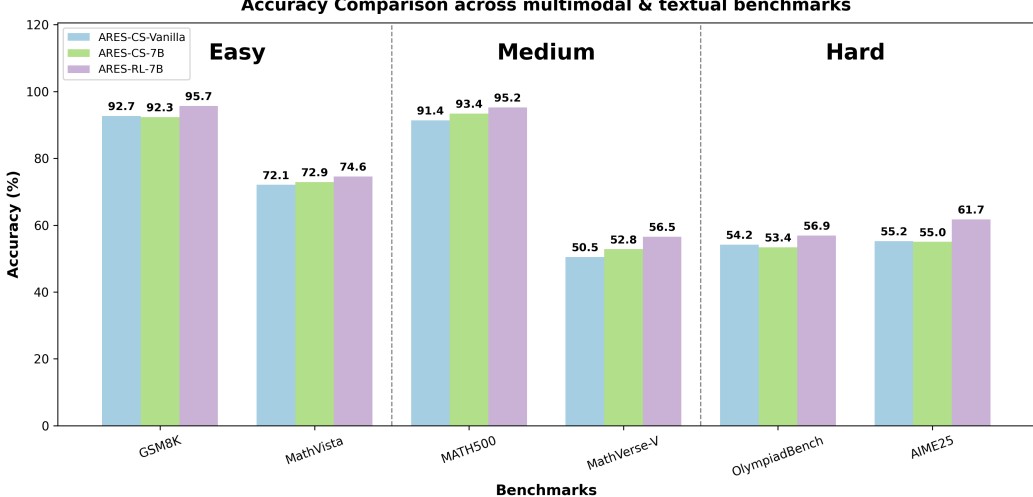

Figure 8: **Accuracy comparison across multimodal and textual benchmarks.** Accuracy of three model variants (ARES-CS-Vanilla, ARES-CS-7B, and ARES-RL-7B) on six benchmarks. Results are grouped into three difficulty levels (Easy, Medium, Hard). RL fine-tuning consistently improves accuracy across all benchmarks.

## K   VISUALIZATION RESULTS

In this subsection, we provide additional visualization results to complement the quantitative findings in the main text. Figures 8 and 9 illustrate the accuracy and response length dynamics across different benchmarks and difficulty levels.

As shown in Figure 8, RL fine-tuning (`ARES-RL-7B`) consistently improves accuracy over both cold-start variants (`ARES-CS-Vanilla` and `ARES-CS-7B`) on all benchmarks. The gains are evident across easy, medium, and hard categories, demonstrating that our proposed reinforcement learning approach enhances reasoning robustness without sacrificing performance on simpler tasks. This confirms the effectiveness of incorporating entropy-aware objectives in stabilizing training and improving generalization.

Figure 9 further analyzes the average response length. We observe that RL fine-tuning reduces response length on easier datasets (e.g., GSM8K, MathVista), reflecting improved reasoning efficiency by eliminating unnecessary verbose reasoning. On more challenging benchmarks (e.g., OlympiadBench, AIME25), the response length of `ARES-RL-7B` increases compared to cold-start models, indicating that the model adaptively allocates more reasoning steps when required. This adaptive behavior—shortening trivial reasoning while allowing deeper exploration on difficult tasks—highlights the benefit of our entropy-shaped reward design.

Together, these visualizations provide clear evidence that our RL-based framework not only improves accuracy but also adapts reasoning length to problem difficulty, thereby balancing efficiency and effectiveness in multimodal reasoning tasks.

## L   WHY HIGH–ENTROPY TOKENS PREDICT REASONING RESPONSE LENGTH

Given input $x$, the policy $\pi_\theta$ generates a response $o = (o_1, \ldots, o_L)$ with random stopping time $\tau = L$. Let $h_t = (x, o_{1:t-1})$ be the history at step $t$ and define the (possibly top-$k$ normalized) token entropy

$$H_t = -\sum_v p_\theta(v \mid h_t) \log p_\theta(v \mid h_t). \tag{11}$$

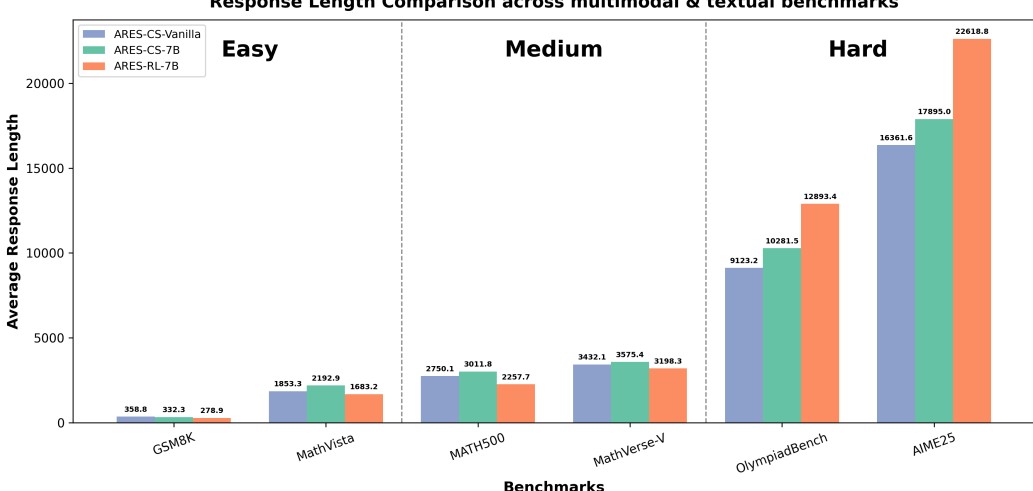

Figure 9: **Response length comparison across multimodal and textual benchmarks.** We report the average number of generated tokens for three model variants (ARES-CS-Vanilla, ARES-CS-7B, and ARES-RL-7B). RL training consistently reduces response length on most benchmarks, indicating improved reasoning efficiency. In contrast, response length increases on the most challenging datasets—AIME25 (textual) and OlympiadBench (multimodal)—highlighting the adaptive behavior of our RL approach: trimming unnecessary reasoning on easy problems while encouraging deeper exploration on difficult ones.

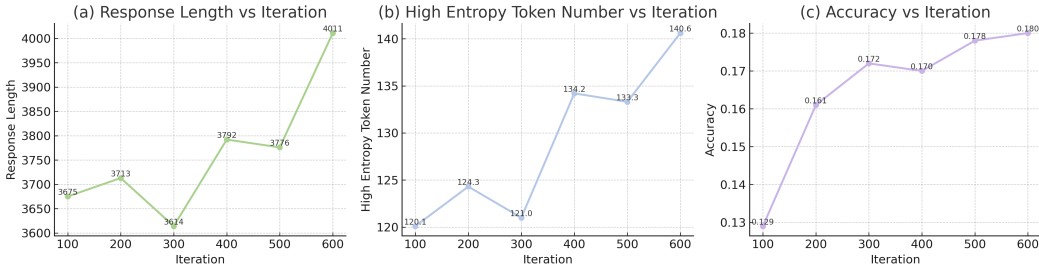

Figure 10: Training dynamics of ValLine GRPO on Coldstart Model: (a) average response length, (b) number of high-entropy tokens, and (c) accuracy, all measured across iterations. The trends indicate that the growth in high-entropy tokens is closely aligned with increases in response length and accuracy.

For a window size $w$ and threshold $\theta$, define window entropy and a high–entropy indicator

$$\bar{H}_{t:w} \;=\; \frac{1}{w} \sum_{\tau=t}^{t+w-1} H_\tau, \qquad \mathbb{I}_t^{\text{HE}} \;=\; \mathbb{I}\{\bar{H}_{t:w} \geq \theta\}, \tag{12}$$

and let the (validated) high–entropy count be

$$N_{\text{HE}} \;=\; \sum_{t=1}^{L} \mathbb{I}_t^{\text{HE}}. \tag{13}$$

Our goal is to relate $L$ to $N_{\text{HE}}$ (or the total high–entropy residence time).

**A two–state latent process.** Assume an interpretable latent state $S_t \in \{\mathsf{R}, \mathsf{V}\}$ with: (i) *Reasoning* ($\mathsf{R}$): exploratory/high–entropy; (ii) *Verbatim* ($\mathsf{V}$): declarative/low–entropy. There exist constants $H_\mathsf{R} > H_\mathsf{V}$ and a threshold $\theta \in (H_\mathsf{V}, H_\mathsf{R})$ such that the detector has bounded error

$$\Pr(\mathbb{I}_t^{\text{HE}} = 1 \mid S_t = \mathsf{R}) \geq 1 - \alpha, \qquad \Pr(\mathbb{I}_t^{\text{HE}} = 0 \mid S_t = \mathsf{V}) \geq 1 - \beta, \tag{14}$$

with $\alpha, \beta \in [0, 1)$. Let the $\mathsf{R} \to \mathsf{V}$ transition probability be $q \in (0, 1]$, and let the answer–emitting hazard in $\mathsf{V}$ be $h \in (0, 1]$ (stopping can only occur in $\mathsf{V}$).

**Theorem 1 (Linear relation with reasoning residence).** Let $T_{\mathsf{R}} = \sum_{t=1}^{\tau} \mathbb{I}\{S_t = \mathsf{R}\}$ be the total time spent in the reasoning state. Then there exist $a, b > 0$ such that

$$\mathbb{E}[L] \;=\; a \;+\; b\,\mathbb{E}[T_{\mathsf{R}}]. \tag{15}$$

Moreover, under equation 14 there exist constants $c_1, c_2 > 0$ for which

$$c_1\,\mathbb{E}[N_{\mathrm{HE}}] \;\leq\; \mathbb{E}[T_{\mathsf{R}}] \;\leq\; c_2\,\mathbb{E}[N_{\mathrm{HE}}], \tag{16}$$

hence

$$\boxed{\mathbb{E}[L] \;=\; a' \;+\; b'\,\mathbb{E}[N_{\mathrm{HE}}]} \quad \text{for some } a', b' > 0. \tag{17}$$

*Proof sketch.* Within $\mathsf{R}$, the process cannot stop; each reasoning excursion has geometric length with mean $1/q$. Within $\mathsf{V}$, the segment ends with geometric stopping hazard $h$ (mean $1/h$). The total length is a renewal sum of i.i.d. excursions plus a terminal $\mathsf{V}$ segment, yielding equation 15. Bounded detector errors map $T_{\mathsf{R}}$ to $N_{\mathrm{HE}}$, proving equation 17. □

**A stopping–time view via entropy–dependent hazard.** Assume the probability of emitting the final answer at step $t$ is a non–increasing function of entropy,

$$\Pr(\text{stop at } t \mid h_t) \;=\; \lambda(H_t), \qquad \lambda'(H) \leq 0. \tag{18}$$

Then larger entropy sequences stochastically dominate the stopping time.

**Theorem 2 (High entropy delays stopping).** For two trajectories with entropy paths $\{H_t^{(1)}\}$ and $\{H_t^{(2)}\}$ such that $H_t^{(1)} \leq H_t^{(2)}$ for all $t$, one has $\mathbb{E}[L^{(1)}] \leq \mathbb{E}[L^{(2)}]$. Moreover, for any threshold $\theta$,

$$\mathbb{E}[L] \;\geq\; \mathbb{E}\left[\sum_{t=1}^{\tau} \frac{\mathbb{I}\{H_t \geq \theta\}}{\lambda(\theta)}\right] \;\geq\; \frac{1}{\lambda(\theta)}\,\mathbb{E}[N_{\mathrm{HE}}]. \tag{19}$$

*Proof sketch.* By stochastic ordering under equation 18, higher entropy lowers the instantaneous hazard and yields larger stopping times in the convex order, implying the expectation inequality. Since $\lambda(H) \leq \lambda(\theta)$ whenever $H \geq \theta$, each high–entropy step contributes at least $1/\lambda(\theta)$ units in expectation; summing gives equation 19. □

**Information–theoretic lower bound.** Suppose answering with error $\varepsilon$ over a label set of size $M$ requires mutual information

$$I^{\dagger} \;\triangleq\; \log M - h_2(\varepsilon) - \varepsilon \log(M - 1) \tag{20}$$

an (by Fano's inequality). Let $i_t \equiv I(A; o_t \mid h_{t-1})$ be per–step information about the (correct) answer $A$. By data processing, $i_t \leq H_t$. Empirically, non–reasoning steps convey negligible information: there exists $\eta \ll 1$ such that $i_t \leq \eta$ when $H_t < \theta$. Let $\iota_\theta \equiv \max_{H \geq \theta} i(H)$. Then

$$I^{\dagger} \;\leq\; \sum_{t=1}^{\tau} i_t \;\leq\; \eta\,(L - N_{\mathrm{HE}}) \;+\; \iota_\theta\, N_{\mathrm{HE}} \;\Rightarrow\; N_{\mathrm{HE}} \;\geq\; \frac{I^{\dagger} - \eta\,\mathbb{E}[L]}{\iota_\theta - \eta}, \tag{21}$$

which rearranges to a linear lower bound of $\mathbb{E}[L]$ in terms of $\mathbb{E}[N_{\mathrm{HE}}]$.

**Synthesis and testable predictions.** The renewal argument (Theorem 1), the hazard view (Theorem 2), and the information budget bound jointly imply a *stable, monotone, near–linear* relationship between reasoning length and high–entropy activity:

$$\mathbb{E}[L] \;\approx\; \alpha \;+\; \beta\,\mathbb{E}[N_{\mathrm{HE}}], \qquad \beta > 0. \tag{22}$$

This yields concrete diagnostics: (i) $\mathbb{E}[L \mid N_{\mathrm{HE}}]$ is monotone increasing; (ii) the slope $\beta$ correlates with the mean reasoning–excursion duration $1/q$; (iii) the estimated stopping hazard $\hat{\lambda}(H)$ is decreasing; and (iv) the per–step mutual information $i_t$ concentrates within high–entropy windows. As shown in Figure 10, the number of high-entropy tokens grows approximately in tandem with response length and accuracy, validating entropy as a principled proxy for reasoning effort. Yet, this relationship is not uniform across all problems: as we show next, the way entropy and reasoning length contribute to accuracy differs substantially between easy and hard instances.

## M    WHY KL LOSS IS A VALID *Thinking Budget*

We show that the KL loss used in AEPO is mathematically equivalent to enforcing a *budget* on the policy's deviation from the reference model, and that such a budget provably controls the expected "thinking" cost (e.g., reasoning length $L$ or entropy-based cost $C_{\text{ent}}$).

For a prompt $x$, let $\pi_\theta(\cdot \mid x)$ be the policy over full responses $o \in \mathcal{Y}$, $\pi_{\text{ref}}(\cdot \mid x)$ the frozen reference, and $r(x, o)$ a task reward. Let $c(x, o) \geq 0$ be a *thinking cost* (e.g., response length, or high-entropy window cost), and define the shaped reward $\tilde{r}(x, o) = r(x, o) - \lambda c(x, o)$ with $\lambda \geq 0$. Denote the (pathwise) KL divergence $D_{\text{KL}}\big(\pi_\theta(\cdot \mid x) \,\|\, \pi_{\text{ref}}(\cdot \mid x)\big)$.

**Constrained formulation.** We pose the *budgeted* policy optimization:

$$\max_{\pi_\theta} \ \mathbb{E}_{x \sim \mathcal{D}}\Big[\mathbb{E}_{o \sim \pi_\theta(\cdot \mid x)}[\tilde{r}(x, o)]\Big] \quad \text{s.t.} \quad \mathbb{E}_{x \sim \mathcal{D}}\Big[D_{\text{KL}}\big(\pi_\theta(\cdot \mid x) \,\|\, \pi_{\text{ref}}(\cdot \mid x)\big)\Big] \leq \delta, \qquad (23)$$

where $\delta > 0$ is the *thinking budget*. The feasible set is convex because $D_{\text{KL}}(\cdot \| \pi_{\text{ref}})$ is convex in its first argument.

**Lemma 2** (Strong duality and KL-as-budget). *Problem equation 23 admits strong duality. Its Lagrangian is*

$$\mathcal{L}(\pi, \kappa) = \mathbb{E}_x \mathbb{E}_{o \sim \pi(\cdot \mid x)}[\tilde{r}(x, o)] \ - \ \kappa \mathbb{E}_x\Big[D_{\text{KL}}\big(\pi(\cdot \mid x) \,\|\, \pi_{\text{ref}}(\cdot \mid x)\big) - \delta\Big], \qquad \kappa \geq 0, \quad (24)$$

*and the optimal policy for fixed $x$ has the exponential-tilt form*

$$\pi^*(o \mid x) \ \propto \ \pi_{\text{ref}}(o \mid x) \ \exp\Big(\tfrac{1}{\kappa} \tilde{r}(x, o)\Big). \qquad (25)$$

*Moreover, at the dual optimum $\kappa^*$ the budget is* active *unless $\kappa^*$ hits its boundary:* $\mathbb{E}_x D_{\text{KL}}\big(\pi^*(\cdot \mid x) \,\|\, \pi_{\text{ref}}(\cdot \mid x)\big) = \delta$.

*Proof.* The objective is linear in $\pi$ and the constraint set is convex with nonempty interior (Slater's condition holds by taking $\pi = \pi_{\text{ref}}$). Thus strong duality holds. Optimizing $\mathcal{L}$ over $\pi(\cdot \mid x)$ under the simplex constraint yields equation 25 via standard exponential-family calculus. Complementary slackness gives the budget activity. $\qquad\square$

**Token factorization and actor-only KL loss.**    For auto-regressive policies, by the chain rule,

$$D_{\text{KL}}\big(\pi_\theta(\cdot \mid x) \,\|\, \pi_{\text{ref}}(\cdot \mid x)\big) = \mathbb{E}_{o \sim \pi_\theta}\Big[\sum_{t=1}^{|o|} D_{\text{KL}}\big(\pi_\theta(\cdot \mid s_t) \,\|\, \pi_{\text{ref}}(\cdot \mid s_t)\big)\Big], \qquad (26)$$

with $s_t = (x, o_{1:t-1})$. Hence a *per-token* KL loss in the actor objective

$$\kappa \cdot \frac{1}{|o|} \sum_t \beta_t \, D_{\text{KL}}\big(\pi_\theta(\cdot \mid s_t) \,\|\, \pi_{\text{ref}}(\cdot \mid s_t)\big)$$

is precisely a discretization of the dual term in equation 23; weights $\beta_t \in (0, \infty)$ allow token-adaptive emphasis (e.g., relaxing inside validated high-entropy windows). Because this term *does not* enter the advantage, it controls deviation magnitude independently of the task/entropy signals that decide *when* to explore.

**Budget tracking via dual updates.**    Let $\widehat{\text{KL}}$ be a moving-average estimate of the left-hand side of equation 23. The multiplicative update

$$\kappa \ \leftarrow \ \text{clip}\Big(\kappa\big(1 + \alpha_\kappa(\widehat{\text{KL}}/\delta - 1)\big), \, \kappa_{\min}, \kappa_{\max}\Big) \qquad (27)$$

is a stochastic dual ascent that drives $\widehat{\text{KL}} \to \delta$. Under standard Robbins–Monro conditions on $\alpha_\kappa$ and bounded variance, $\kappa_t \to \kappa^*$ a.s., and the primal iterates satisfy the budget asymptotically. Thus the actor-only KL loss with controller implements an *operational thinking budget*.

**From KL budget to *thinking* budget.** We now show that bounding KL controls the expected thinking cost (e.g., $c = L$ or $c = C_{\text{ent}}$).

**Lemma 3** (Pinsker-type bound). *For any bounded $f : \mathcal{Y} \to \mathbb{R}$ with $\|f\|_\infty \leq M$,*

$$\left| \mathbb{E}_{\pi_\theta}[f] - \mathbb{E}_{\pi_{\text{ref}}}[f] \right| \leq M \sqrt{2 D_{\text{KL}}(\pi_\theta \| \pi_{\text{ref}})}. \tag{28}$$

Thus if $f$ is the (clipped) reasoning length or normalized entropy cost, a *global* KL budget $\delta$ implies a corresponding bound on the change of its expectation.

**Theorem 1** (Donsker–Varadhan control of moment budgets). *For any $\eta > 0$ and measurable $f$,*

$$\mathbb{E}_{\pi_\theta}[f] \leq \frac{1}{\eta} \log \mathbb{E}_{\pi_{\text{ref}}}\left[e^{\eta f}\right] + \frac{1}{\eta} D_{\text{KL}}(\pi_\theta \| \pi_{\text{ref}}). \tag{29}$$

*Consequently, under the budget $D_{\text{KL}} \leq \delta$,*

$$\mathbb{E}_{\pi_\theta}[f] \leq \inf_{\eta > 0} \left\{ \frac{1}{\eta} \log \mathbb{E}_{\pi_{\text{ref}}}[e^{\eta f}] + \frac{\delta}{\eta} \right\}. \tag{30}$$

*Proof.* equation 29 is the Donsker–Varadhan variational inequality obtained by upper-bounding the log-moment generating function via relative entropy. $\square$

Taking $f = c(x, o)$ (e.g., $L$ or $C_{\text{ent}}$) shows that a KL budget *upper-bounds* the expected thinking cost relative to the reference through the reference MGF, i.e., the policy cannot arbitrarily increase reasoning length or entropy cost without paying KL.

**Mirror-descent / trust-region view.** Maximizing $\mathbb{E}[\tilde{r}]$ under a *local* KL ball $D_{\text{KL}}(\pi \| \pi_{\text{old}}) \leq \delta$ yields the mirror-descent (natural-gradient) update

$$\pi^* \propto \pi_{\text{old}} \exp\left(\tfrac{1}{\kappa} \tilde{r}\right), \quad \text{with } \kappa \asymp \frac{1}{\sqrt{\delta}}, \tag{31}$$

so $\delta$ is precisely the *trust-region radius*: a smaller (larger) budget forces smaller (larger) policy movement and therefore smaller (larger) allowable growth of $c$ by Lemma 3/Theorem 1. In auto-regressive models, equation 26 further identifies the budget with a *tokenwise* sum of deviations, making the KL loss a time–additive *budget meter* of exploration.

**Token-adaptive windows preserve convexity.** Let $\{w_t\}$ be nonnegative weights (e.g., $w_t \in [\rho, 1]$ with $\rho < 1$ inside validated reasoning windows). The weighted budget $\sum_t w_t D_{\text{KL}}(\pi_t \| \pi_t^{\text{ref}})$ remains convex in $\{\pi_t\}$ and admits the same dual treatment; hence the actor-only term $\kappa \sum_t w_t \, \text{kld}_t$ is still a valid Lagrangian penalty for a *window-relaxed* KL budget. This realizes the intuition: *"relax KL where we intend to think"* without breaking the budgeting semantics.

(i) Strong duality turns the KL loss into the Lagrange multiplier of the reference-deviation budget; (ii) dual updates equation 27 track the target $\delta$; (iii) information inequalities (Pinsker, Donsker–Varadhan) translate a KL budget into explicit upper bounds on the expected thinking cost. Therefore, the actor-only KL loss in AEPO is not merely a regularizer: it is a *principled and operational thinking budget*.

# N   FISHER–GEOMETRY JUSTIFICATION OF AEPO

This subsection formalizes why the proposed *Adaptive–Entropy Policy Optimization* (AEPO) is principled from the viewpoint of information geometry. We show that AEPO is equivalent to a *token–reweighted natural–gradient update under a per–difficulty trust region*, and that the entropy–augmented advantage and token–adaptive KL together maximize improvement in the directions that matter for reasoning.

**Setup.** Given a prompt $x$, the auto–regressive policy $\pi_\theta$ generates a response $o = (o_{1:L})$ with states $s_t = (x, o_{<t})$. Let $r(x, o)$ be the task reward and $C_{\text{ent}}(x, o)$ the entropy–window cost (Section 2.1). For a difficulty bucket $d = d(x)$, AEPO solves the following constrained problem:

$$\max_{\pi_\theta} \quad \mathbb{E}_{x\sim\mathcal{D}} \, \mathbb{E}_{o\sim\pi_\theta(\cdot|x)} \Big[ r(x, o) \; - \; \lambda_{d(x)} \, C_{\text{ent}}(x, o) \Big]$$

$$\text{s.t.} \quad \mathbb{E}_{x\sim\mathcal{D}} \, \mathbb{E}_{o\sim\pi_\theta(\cdot|x)} \left[ \frac{1}{L(o)} \sum_{t=1}^{L(o)} \beta_{d(x),t} \, D_{\text{KL}}\Big(\pi_\theta(\cdot \mid s_t) \, \big\| \, \pi_{\text{ref}}(\cdot \mid s_t)\Big) \right] \; \leq \; \delta_{d(x)} \,. \tag{32}$$

Here $L(o)$ denotes the response length and $s_t = (x, o_{<t})$. The token weight is $\beta_{d,t} = \beta_d \, \rho_t$, where $\rho_t \in (0, 1)$ if $t$ lies in a validated high–entropy window and $\rho_t = 1$ otherwise.

**KL as a Fisher quadratic.** Let $F_t$ denote the token–wise Fisher information matrix under the reference policy $\pi_{\text{ref}}$:

$$F_t = \mathbb{E}_{a\sim\pi_{\text{ref}}(\cdot|s_t)} \big[ \nabla \log \pi_{\text{ref}}(a \mid s_t) \, \nabla \log \pi_{\text{ref}}(a \mid s_t)^\top \big].$$

For a small parameter displacement $\Delta\theta$, the per–token KL admits the standard second–order approximation

$$D_{\text{KL}}\big(\pi_{\theta+\Delta\theta}(\cdot \mid s_t) \, \big\| \, \pi_{\text{ref}}(\cdot \mid s_t)\big) \; = \; \tfrac{1}{2} \Delta\theta^\top F_t \, \Delta\theta \; + \; o(\|\Delta\theta\|^2). \tag{33}$$

**Lemma 4** (Weighted Fisher trust region). *Under the approximation equation 33, the KL constraint in equation 32 is equivalent to the quadratic trust region $\tfrac{1}{2} \Delta\theta^\top F_\beta \Delta\theta \leq \delta_d$, with the token–reweighted Fisher*

$$F_\beta \triangleq \frac{1}{L} \sum_{t=1}^{L} \beta_{d,t} \, F_t, \qquad \beta_{d,t} = \beta_d \, \rho_t, \;\; \rho_t \in (0, 1]. \tag{34}$$

*Proof.* Average equation 33 over $t$ with weights $\beta_{d,t}$ and use linearity. $\qquad\square$

**Dual/Lagrangian form and natural gradient.** Introducing a Lagrange multiplier $\kappa_d \geq 0$, the inner (fixed–$d$) problem becomes

$$\max_{\Delta\theta} \; g^\top \Delta\theta \; - \; \tfrac{\kappa_d}{2} \, \Delta\theta^\top F_\beta \, \Delta\theta, \qquad g \triangleq \nabla_\theta \, \mathbb{E}\big[ r(x, o) - \lambda_d \, C_{\text{ent}}(x, o) \big], \tag{35}$$

whose maximizer is the natural–gradient step

$$\boxed{\; \Delta\theta^\star \; = \; \kappa_d^{-1} \, F_\beta^{-1} \, g \;} \tag{36}$$

**Proposition 2** (One–step improvement bound). *Under equation 33, the first–order improvement at $\Delta\theta^\star$ satisfies*

$$\Delta J \triangleq g^\top \Delta\theta^\star \; - \; \tfrac{\kappa_d}{2} \, \Delta\theta^{\star\top} F_\beta \Delta\theta^\star \; = \; \frac{1}{2\kappa_d} \, g^\top F_\beta^{-1} g. \tag{37}$$

*Proof.* Plug equation 36 into equation 35. $\qquad\square$

**Why token–adaptive relaxation helps.** Inside validated reasoning windows we choose $\rho_t < 1$, which reduces the local curvature contribution in $F_\beta$ on those tokens. Simultaneously, the entropy–augmented advantage increases the corresponding components of $g$ only where $\bar{H}_{t:w} \geq \vartheta$. Because the gain equation 37 increases when (i) $g$ has larger energy and (ii) the metric penalty $F_\beta$ is smaller in the same directions, AEPO aligns a boosted gradient with a relaxed Fisher metric on reasoning windows.

**Corollary 1** (Directional benefit of window relaxation). *Let $u$ be a unit vector supported on window tokens. If $\rho_t$ is reduced on these tokens, then $u^\top F_\beta u$ decreases while $u^\top g$ increases under entropy shaping. Hence the directional contribution $(u^\top g)^2 / (2\kappa_d \, u^\top F_\beta u)$ to equation 37 strictly increases, improving sample efficiency on reasoning segments.*

**Why use *non–window* KL for control.** The KL controller adjusts $\kappa_d$ to match the per–bucket target $\delta_d$. If we fed the controller with the already–relaxed (window–weighted) KL, the true natural length $\frac{1}{2}\Delta\theta^\top(\frac{1}{L}\sum_t F_t)\Delta\theta$ could be under–estimated, risking drift. Using a non–window (or down–weighted–window) control signal $\mathrm{KL}_{\mathrm{ctrl}}$ stabilizes the global trust region while keeping local relaxation intact.

**Per–bucket decoupling: shape vs. scale.** The token weights $\beta_{d,t}$ determine the *shape* of the metric $F_\beta$ (where to allow deviation), whereas the multiplier $\kappa_d$ determines its *scale* (how much to deviate) for each difficulty bucket. This yields a well–conditioned, two–time–scale scheme: slow adaptation of $\kappa_d$ (budget tracking) and fast ascent in equation 35.

AEPO corresponds to a per–difficulty, token–reweighted natural–gradient update

$$\Delta\theta^\star \;=\; \kappa_d^{-1}\Big(\tfrac{1}{L}\sum_t \beta_{d,t}\,F_t\Big)^{-1}\underbrace{\nabla_\theta\mathbb{E}\big[r - \lambda_d\,C_{\mathrm{ent}}\big]}_{\text{entropy–shaped } g},$$

with $\beta_{d,t} \;=\; \beta_d\rho_t$ relaxing KL strictly within high–entropy windows and $\kappa_d$ controlled by a non–window KL signal. Proposition 2 shows this maximizes $g^\top F_\beta^{-1}g/(2\kappa_d)$ by (i) amplifying $g$ where the model is genuinely reasoning and (ii) reducing curvature along the same directions, while a stable bucket–wise trust region ensures controlled deviation from $\pi_{\mathrm{ref}}$.

## O  CASE STUDY

QUESTION

As shown in the figure, in the right triangle one acute angle is $60°$ and the side opposite this angle has length $6\sqrt{3}$. Find the hypotenuse $y$.

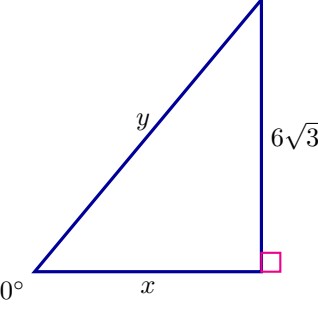

**Answer:** $\boxed{12}$

Tokens: 243

OUR ADAPTIVE MODEL SOLUTION FOR EASY QUESTION

Use the sine ratio for the $60°$ angle:

$$\sin 60° = \frac{\text{opposite}}{\text{hypotenuse}} = \frac{6\sqrt{3}}{y}.$$

Since $\sin 60° = \frac{\sqrt{3}}{2}$, we have

$$\frac{\sqrt{3}}{2} = \frac{6\sqrt{3}}{y} \quad\Longrightarrow\quad y = \frac{6\sqrt{3}\cdot 2}{\sqrt{3}} = 12.$$

$$\boxed{y = 12}$$

QUESTION

The sum of the circumferences of circles $H$, $J$, and $K$ is $56\pi$ units. Their radii are marked as $r_H = x$, $r_J = 2x$, $r_K = 4x$, and the three circles are pairwise tangent externally. Find the distance $KJ$ between the centers of circles $K$ and $J$.

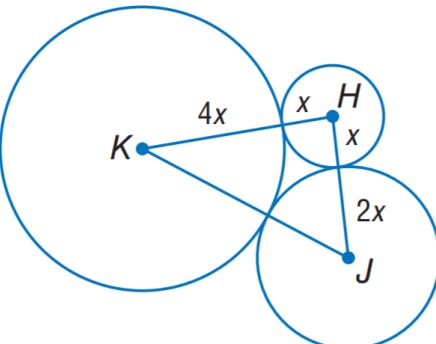

**Tokens: 4173**

OUR ADAPTIVE MODEL SOLUTION FOR HARD QUESTION

**First**, recall that the circumference of a circle is $2\pi r$. The problem states that the sum of the circumferences of circles $H, J, K$ is $56\pi$, so

$$2\pi(r_H + r_J + r_K) = 56\pi \implies r_H + r_J + r_K = 28.$$

**At this point**, I examined the diagram and noticed the labels $4x$, $x$, $2x$. I hesitated: are these radii, or are they distances between centers? If interpreted as center distances, then

$$KH = 4x, \quad HJ = x, \quad JK = 2x,$$

but this quickly fails because $x + 2x = 3x < 4x$, violating the triangle inequality. Therefore, this interpretation cannot be correct.
**Next**, I considered that these labels instead denote the radii:

$$r_H = x, \quad r_J = 2x, \quad r_K = 4x.$$

Substituting into $r_H + r_J + r_K = 28$ yields

$$x + 2x + 4x = 28 \implies 7x = 28 \implies x = 4.$$

**However**, doubts persisted. For instance, if one misreads $JK$ as $2x$, then $KJ$ would be $8$, which contradicts the tangent-circle rule requiring

$$KJ = r_K + r_J = 4x + 2x = 6x = 24.$$

... Here I went back and forth, testing whether the labels might be mixed as radii or distances, but each alternative led to inconsistencies. For example, assuming $KH = 4x$ as a distance forces $r_K = 3x$ if tangent to $r_H = x$, yet then $HJ$ should be $3x$ not $x$, another contradiction. These cycles of reasoning strongly suggested that the only consistent interpretation is that the labels $4x$, $x$, $2x$ represent the radii themselves.
**Therefore**, with $x = 4$ we compute the actual radii:

$$r_K = 16, \quad r_H = 4, \quad r_J = 8.$$

Since circles $K$ and $J$ are externally tangent, the distance between their centers is

$$KJ = r_K + r_J = 16 + 8 = 24.$$

**Finally**, check consistency:

$$2\pi(r_H + r_J + r_K) = 2\pi(4 + 8 + 16) = 2\pi(28) = 56\pi,$$

matching the given condition exactly.

$$\boxed{24}$$

QUESTION

In $\odot W$, arcs $\overset{\frown}{RS}$ and $\overset{\frown}{TV}$ are congruent. Find the length of $RS$.

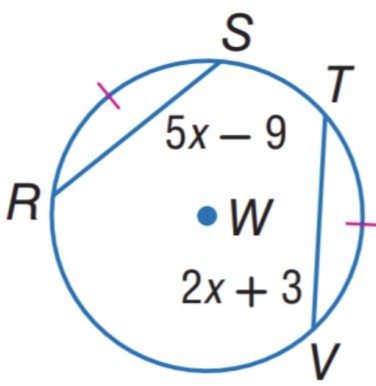

QUESTION WITH APPROPRIATE NUMBER OF HIGH-ENTROPY TOKENS

**difficulty:** Easy      **Gen_high_entropy_token:** 20 **Threshold(easy):** 28
**Entropy reward:** 0.0 **acc reward:** 1.0      **overall reward:** 1.0

**Okay, let's see.** I need to find the length of $RS$ in circle $\odot W$. The problem says the arcs $RS$ and $TV$ are congruent.

**First, recall** that in a circle, congruent arcs correspond to congruent chords. **Therefore**, if arc $RS \cong$ arc $TV$, then chord $RS \cong$ chord $TV$.

The problem gives:
$$RS = 5x - 9, \quad TV = 2x + 3.$$

**So we set them equal:**
$$5x - 9 = 2x + 3.$$

**Solving:**
$$3x - 9 = 3 \implies 3x = 12 \implies x = 4.$$

**Now check:**
$$RS = 5(4) - 9 = 11, \quad TV = 2(4) + 3 = 11.$$

**Wait, let me double-check.** Are RS and TV the correct chords? Arc $RS$ corresponds to chord $RS$, arc $TV$ to chord $TV$. Since arcs are congruent, chords must be equal. Yes, consistent.

**Thus**, the required length is
$$\boxed{11}.$$

QUESTION

In $\triangle ABC$, the side lengths are $a = 14$, $b = 48$, and $c = 50$. Find $\cos B$.

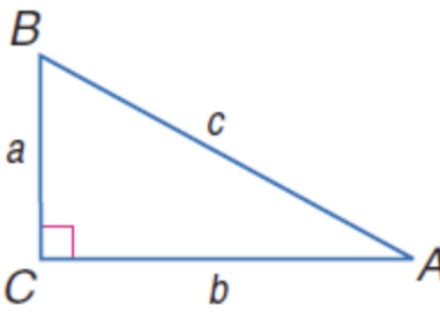

QUESTION WITH EXCESS HIGH-ENTROPY TOKENS

**difficulty:** Easy **Gen_high_entropy_token:** 51 **Threshold(easy):** 28
**Entropy reward:** -0.5 **acc reward:** 1.0 **overall reward:** 0.5

**Okay, so** I need to find $\cos B$ given a triangle with sides $a = 14$, $b = 48$, and $c = 50$. **Let me recall** the Law of Cosines:

$$c^2 = a^2 + b^2 - 2ab\cos C.$$

**First**, label the triangle properly. Side $a$ is opposite angle $A$, $b$ opposite $B$, and $c$ opposite $C$. The diagram shows $\angle C = 90°$. **Therefore**, $c = 50$ must be the hypotenuse, and $a = 14$, $b = 48$ are the legs.

**Now**, angle $B$ is at vertex $B$, between sides $BC$ ($a = 14$) and $AB$ ($c = 50$). The opposite side is $AC$ ($b = 48$). **Thus**, by definition:

$$\cos B = \frac{\text{adjacent}}{\text{hypotenuse}} = \frac{a}{c} = \frac{14}{50} = \frac{7}{25}.$$

**But let me confirm** with the Law of Cosines. For angle $B$:

$$\cos B = \frac{a^2 + c^2 - b^2}{2ac}.$$

**Substitute carefully:**

$$\cos B = \frac{14^2 + 50^2 - 48^2}{2 \cdot 14 \cdot 50} = \frac{196 + 2500 - 2304}{1400} = \frac{392}{1400} = \frac{7}{25}.$$

**Wait, let me double-check.** If $a = 14$, $b = 48$, $c = 50$, then indeed $14^2 + 48^2 = 196 + 2304 = 2500 = 50^2$, confirming it is a right triangle. **Hence** the direct trigonometric ratio already suffices.

... (several "wait" moments, checking whether sides match standard notation, omitted for brevity) ...

**Finally**, both direct definition and Law of Cosines give the same result. **Therefore**, the value is consistent:

$$\cos B = \frac{7}{25} \approx 0.28.$$

$$\boxed{\frac{7}{25}}$$

QUESTION

Find $x$.

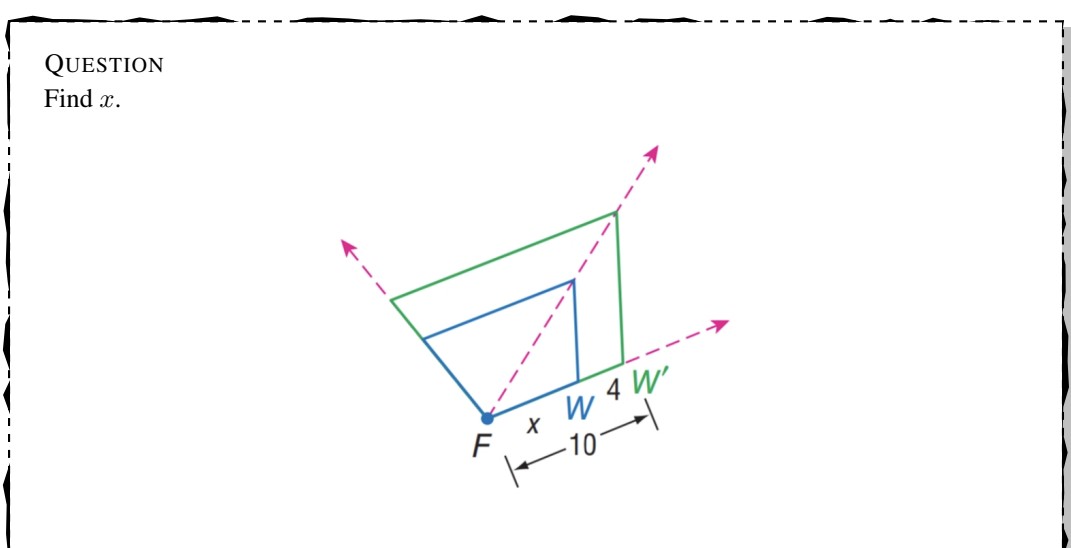

MEDIUM QUESTION WITH EXCESS HIGH-ENTROPY TOKENS (INCORRECT ANSWER)

**difficulty:** Medium     **Gen_high_entropy_token:** 84 **Threshold(medium):** 55
**Entropy reward:** 0.38 **acc reward:** 0.0                    **overall reward:** 0.38

**Okay, let's try** to figure out how to find $x$. The problem involves a diagram with a green square and a blue square connected. There is a point $F$ with a segment of length $10$, and another segment labeled $4$.

**Wait, maybe** the diagram is a cube net, since dashed lines often indicate hidden edges. If $FW = 10$ and $WW' = 4$, perhaps $x$ is the distance from $F$ to $W'$.

**Alternatively**, maybe this is related to vectors or a triangle. If $x$, $10$, and $4$ form sides of a right triangle, then

$$x^2 = 10^2 + 4^2 = 116, \quad x = \sqrt{116} = 2\sqrt{29}.$$

**But hold on**, maybe the 10 and 4 are not perpendicular. Then by the Law of Cosines:

$$x^2 = 10^2 + 4^2 - 2 \cdot 10 \cdot 4 \cos\theta,$$

but without $\theta$, this is indeterminate.

**Next**, let me try coordinates. Suppose $F = (0, 0, 0)$. **If** $W = (10, 0, 0)$ and $W' = (10, 4, 0)$, then
$$FW' = \sqrt{10^2 + 4^2} = \sqrt{116}.$$
**Alternatively**, if $W' = (10, 0, 4)$, the distance is also $\sqrt{116}$.

... (several further "wait" moments about whether this is a parallelogram, trapezoid, or 3D projection are omitted here) ...

**Another thought**: perhaps FW and WW' are adjacent edges of a cube net, so $x$ is the straight-line distance from $F$ to $W'$. **Therefore**, again,

$$x = \sqrt{10^2 + 4^2} = 2\sqrt{29}.$$

**Yet**, could it be colinear? If so, $x = 10 + 4 = 14$. But that seems inconsistent with the diagram.
**Maybe** it's a right triangle hidden in the net. If FW is one leg, WW' another, then $x$ is the hypotenuse.
**Finally**, after multiple conflicting attempts, the reasoning converges to

$$x = 2\sqrt{29}.$$

QUESTION
Find y.

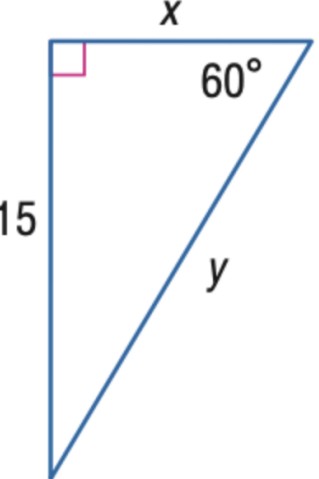

MEDIUM QUESTION WITH FEW HIGH-ENTROPY TOKENS (INCORRECT ANSWER)

**difficulty:** Medium      **Gen_high_entropy_token:** 24 **Threshold(medium):** 55
**Entropy reward:** 0.19 **acc reward:** 0.0                **overall reward:** 0.19

**Okay, so** I need to find the value of $y$ in this triangle. The diagram shows a right triangle with a $60°$ angle, a side of length 15, and the hypotenuse labeled $y$.
**Since** it is a right-angled triangle, the other non-right angle must be $30°$.

**In a 30-60-90 triangle**, the sides are in ratio $1 : \sqrt{3} : 2$. The side opposite $30°$ is the shortest.

**If** the side of length 15 is opposite $30°$, then the hypotenuse should be $2 \times 15 = 30$.

**Alternatively**, using $\cos 60° = \frac{1}{2}$:

$$\cos 60° = \frac{15}{y} \quad \Longrightarrow \quad y = 30.$$

**But**, in the problem statement the side labeled 15 might not match this assumption. If it were opposite $60°$, the calculation would change.
... (further checks on labeling are omitted) ...

**Finally**, the reasoning settled on
$$y = 30.$$

