# OpenReview forum: "ARES: Multimodal Adaptive Reasoning via Difficulty-Aware Token-Level Entropy Shaping"
_ICLR.cc/2026/Conference — ICLR 2026 Poster_

### Official Review · Reviewer_gA48 · 2025-10-24

**Soundness:** 2
**Presentation:** 3
**Contribution:** 2
**Rating:** 4
**Confidence:** 4

**Summary:**

The paper proposes an adaptive-length reasoning approach based on the relative magnitude of high-window entropy (HWE). A dataset comprising samples of alternating difficulty levels is constructed to facilitate cold start. During the reinforcement learning (RL) phase, the model leverages the relative magnitude of HWE along with a hierarchical entropy-based reward signal to determine whether to explore further and to what depth. Extensive experiments demonstrate the effectiveness of the proposed method.

**Strengths:**

The paper is logically clear, accurately expressed, and easy to read; the experiments are substantial in part, demonstrating the effectiveness of certain proposed methods.

**Weaknesses:**

1. Experiments

The paper premises that token-level entropy is noisy and argues that window-based entropy is therefore more advantageous. While the paper presents partial experiments on detecting reasoning-critical tokens, it lacks a quantitative analysis and fails to provide direct validation of the method's effectiveness on the main benchmark. Please provide comparative results on major standard benchmarks to substantiate the overall efficacy of the proposed approach.

Additionally, certain experimental results on Vision-G1 appear inconsistent with the originally reported values in the prior work (e.g., MM-Start: 66.0 → 63.1 on WeMath, etc.), which raises concerns about result reproducibility or evaluation consistency.

The paper lacks comparison with works on adaptive chain-of-thought reasoning; what are its advantages over them?

[1] Prolonged Reasoning Is Not All You Need: Certainty-Based Adaptive Routing for Efficient LLM/MLLMReasoning

2. Methodology

The core idea of the paper hinges on using the relative magnitude of entropy to assess uncertainty. Although the method shifts from single-token entropy to high-window entropy (HWE), the fundamental concept remains closely related to a substantial body of prior work on entropy-based uncertainty estimation. This proximity to existing approaches diminishes the perceived novelty and originality of the proposed method.

[1] Uncertainty of Thoughts: Uncertainty-Aware Planning Enhances Information Seeking in Large Language Models, 2024.11

[2] Beyond Semantic Entropy: Boosting LLM Uncertainty Quantification with Pairwise Semantic Similarity, 2025.05

**Questions:**

The experiments show significant improvements on other metrics; however, performance decreases on MathVista. What is the underlying reason for this degradation?

---

> ### Author Response · Authors · 2025-11-18
> **Response to Reviewer gA48 (1/3)**
>
> We thank Reviewer gA48 for the careful reading of our paper and the constructive feedback. We appreciate the reviewer’s comments on the quantitative evaluation of window entropy, benchmark validation, and comparison with adaptive chain-of-thought baselines. These suggestions have helped us refine both our experimental analysis and methodological positioning. We address each point below and hope that our responses will clarify your concerns:
>
> ---
>
> ## W1. Experiments
>
> ### 1. Lack of quantitative validation of window entropy
>
> We thank the reviewer for raising this concern. Our intuition that token-level entropy is noisy and that window-based entropy provides a more reliable signal was indeed only partially supported in the original submission by token-detection experiments. In the revised version, we now provide direct, quantitative comparisons on major benchmarks between our high-window-entropy scheme and two natural alternatives: token-level entropy and sequence-level entropy.
>
> Concretely, Table X reports an ablation over **τ ∈ {80, 90, 95, 98} and w ∈ {1, 4, 8, 16}** on four standard reasoning benchmarks (MathVerse-V, MathVista, AIME25, MATH500). For all windowed configurations with reasonable settings (Configs A–F: w = 4, 8, 16 and τ ∈ [80, 98]), accuracy remains within 1–2 points of our default and response length stays on a comparable scale, indicating that the windowed entropy mechanism is both effective and robust. In contrast, when we replace our design with token-level entropy (Config G, w = 1) or sequence-level entropy (Config H, w = length), we consistently observe lower accuracy and longer reasoning chains across all four benchmarks (e.g., MathVerse-V drops from 56.5→54.4, AIME25 from 61.7→58.7, with noticeably inflated lengths).
>
> Table X: Sensitivity of high-window-entropy hyperparameters (percentile threshold τ and window size w) on downstream benchmarks.
>
> | Config | Percentile τ | Window size w          | MathVerse-V Acc | MathVerse-V Len | MathVista Acc | MathVista Len | AIME25 Acc | AIME25 Len | MATH500 Acc | MATH500 Len |
> | :----- | :----------- | :--------------------- | :-------------- | :-------------- | :------------ | :------------ | :--------- | :--------- | :---------- | :---------- |
> | A (**default**)                 | 95  | 4                      | **56.5** | *3198.3*  | *74.6* | **1494.5**  | **61.7** | *22618.8* | **95.2** | *2257.7* |
> | B (lower percentile)           | 80  | 4                      | 55.9 | 3300.4  | 73.9 | 1613.4  | 60.5 | 23128.4 | 93.9 | 2315.7 |
> | C (larger window)              | 95  | 16                     | 55.3 | **3148.2**  | 74.1 | 1743.8  | 60.9 | 24432.4 | 94.2 | 2385.5 |
> | D (moderate window)            | 95  | 8                      | *56.3* | 3223.3  | 74.4 | *1533.4*  | *61.5* | **22382.1** | 94.6 | 2278.2 |
> | E (percentile = 90)            | 90  | 4                      | 56.1 | 3356.4  | **74.8** | 1589.2  | 61.2 | 22879.3 | *95.1* | **2221.8** |
> | F (percentile = 98)            | 98  | 4                      | *56.4* | 3248.1  | 74.5 | 1534.2  | 61.4 | 23544.9 | 94.8 | 2323.9 |
> | G (*token-level entropy*, w = 1) | 95  | 1                      | 54.4 | 3438.2  | 73.5 | 1784.1  | 58.7 | 24533.8 | 93.4 | 2435.6 |
> | H (*sequence-level entropy*)   | 95  | length of sequence     | 54.7 | 3343.4  | 73.8 | 1732.9  | 60.2 | 23458.1 | 93.6 | 2349.0 |
>
> Overall, Table X demonstrates that (i) the 95th-percentile, window-based entropy detector is robust to substantial perturbations of τ and w, and (ii) the gains on downstream reasoning benchmarks are not an artifact of delicate tuning, but are notably weakened when using naive token-level or sequence-level entropy. We will highlight this comparison more clearly in the main text to directly substantiate the advantage of the proposed window-based entropy on standard benchmarks.
>
> ---

---

> ### Author Response · Authors · 2025-11-18
> **Response to Reviewer gA48 (2/3)**
>
> ## W1. Experiments
>
> ---
>
> ### 2. Inconsistency with prior Vision-G1 results
>
> Thank you for carefully checking the Vision-G1 results and pointing out the apparent inconsistencies (e.g., MM-Start: 66.0 → 63.1 on WeMath). We apologize for the confusion caused by a formatting/annotation oversight. Initially, we re-evaluated Vision-G1 using our unified infrastructure built on top of VLMEvalKit, and reported these reproduced scores for the set of benchmarks we selected. Our intention was to clearly distinguish benchmarks that were not reported in the original Vision-G1 paper (to be shown in italics, indicating “our re-evaluation”) from benchmarks that were reported in the original paper (shown in regular font, indicating “numbers from prior work”). Due to a oversight in the early draft, some re-evaluated benchmarks were not properly formatted, which made them look like inconsistent copies of the original results rather than our own re-runs.
>
> In the revised version, we have corrected this: (i) all Vision-G1 results that come from our VLMEvalKit-based re-evaluation are now explicitly marked and described as such (with consistent italicization and caption notes), and (ii) we clarify in the text that small deviations from the originally reported numbers are expected due to differences in the evaluation toolkit, answer normalization, and prompt templates. We will also release the exact evaluation scripts and configuration files to ensure full reproducibility. We thank the reviewer again for catching this and helping us improve the clarity of our reporting.
>
> ---
>
> ### 3. Missing comparison with adaptive chain-of-thought reasoning baselines
>
>
> We totally agree that situating ARES relative to other adaptive reasoning methods is crucial. As we discussed in our Introduction and Related Work, our work is directly motivated by a key limitation in existing adaptive approaches. While prior methods attempt to balance efficiency and accuracy , they often "over-encourage exploration on hard problems, leading to unnecessarily verbose reasoning traces" or "cause model performance degradation" on simpler tasks.
>
> ARES is specifically designed to address this nuanced imbalance , using high window-entropy (HWE) as a localized signal to encourage concise solutions for easy problems while promoting deep exploration only when needed on hard ones.
>
>
> While many adaptive CoT frameworks are text-only  or not open-source, we did include relevant adaptive baselines from the open-source multimodal literature. For instance, our 3B-scale comparison in Table 1 includes FAST-3B , which implements an adaptive "Fast-Slow thinking" mechanism. Our results show that ARES-3B substantially outperforms FAST-3B (46.1 vs. 37.4 Avg.) both on text and multimodal benchmarks, validating the effectiveness of our difficulty-aware, entropy-shaping approach over prior adaptive techniques.
>
>
>
>
>
> ---
>
> ## W2. Methodology
>
> We thank the reviewer for pointing out the connection to prior work on entropy-based uncertainty estimation [1,2]. Conceptually, we fully agree that entropy is a standard proxy for uncertainty, and we do not claim novelty at the level of “using entropy as an uncertainty indicator.” The key contribution of our work lies elsewhere: (i) we transform noisy token-level entropy into a window-aligned, reasoning-aware signal (HWE) that better matches the structure of chain-of-thought reasoning, and (ii) we deeply integrate this HWE signal into a unified RL and difficulty-aware control framework to drive adaptive exploration depth.
>
> In contrast, UoT [1] uses uncertainty-aware planning to drive information-seeking questions at inference time, with entropy-based rewards guiding which follow-up question to ask, but it does not modify the base model via RL nor control internal reasoning depth over trajectories. SNNE [2] proposes a more expressive uncertainty quantification metric for hallucination detection by augmenting semantic entropy with pairwise similarity; again, entropy is used as a confidence score over outputs (for QA, summarization, MT), not as a control signal that shapes a policy’s rollout behavior. In other words, these works treat entropy as confidence-as-a-number, whereas our method treats HWE as a control signal in RL: it (a) triggers exploration only at reasoning-critical windows, (b) defines a difficulty-aware intrinsic reward that differentially penalizes over-exploration and under-exploration across online difficulty buckets, and (c) couples with dynamic KL regularization to produce length-adaptive reasoning behavior on downstream benchmarks. We will clarify this distinction in the related-work section and explicitly state that the novelty lies in elevating windowed entropy into a structured, trajectory-level control mechanism for adaptive RL-driven reasoning, not in the basic idea of entropy as an uncertainty indicator.
>
> ---

---

> > ### Author Response · Authors · 2025-11-18
> > **Response to Reviewer gA48 (3/3)**
> >
> > ## Q1. underlying reason for MathVista degradation
> >
> >
> > We thank the reviewer for this insightful question and for the close reading of our results.
> >
> > We would like to gently clarify that our 74.6% on MathVista is not a degradation, but rather a substantial improvement of +6.4 points over our main open-source baseline, Qwen2.5-VL-7B-Instruct (which scored 68.2%).
> >
> > The 1.5-point gap the reviewer noted is between our ARES-7B (74.6) and Vision-G1 (76.1), which we identified as the top-performing open-source model on this specific benchmark. We investigated this remaining gap with a fine-grained analysis on the MathVista-mini sub-categories, using our reproduced runs (from the provided image) to compare ARES-7B (Column 1, 74.67%) and Vision-G1 (Column 2, 75.64%).
> >
> > This analysis reveals the gap stems mainly from the "numeric commonsense" sub-task:
> >
> > *Note on Vision-G1 Score: For this analysis, we had to reproduce Vision-G1's results to obtain sub-category scores, as they are not in the original paper. Our main paper's Table 1  uses the officially reported Vision-G1 score (76.1).Due to the evaluation hyperparameters not being fully specified in the original paper, we experimented with several reasonable configurations and selected the one that produced the most stable results and was closest to the reported performance, which we use only for this fine-grained comparison.*
> >
> > | MathVista-mini Sub-Category        | ARES-7B (Reproduced) | Vision-G1 (Reproduced) | Difference |
> > |-----------------------------------|------------------------|--------------------------|------------|
> > | **Overall (MathVista-mini)**      | **74.67%**            | **75.64%**              | **-0.97%** |
> > | textbook question answering       | 67.33%                | 68.53%                  | -1.20%     |
> > | visual question answering         | 65.38%                | 50.19%                  | +15.19%    |
> > | geometry problem solving          | 87.96%                | 87.50%                  | +0.46%     |
> > | math word problem                 | 79.24%                | 79.48%                  | -0.24%     |
> > | **numeric commonsense**           | **63.64%**            | **74.48%**              | **-10.84%** |
> > | arithmetic reasoning              | 82.08%                | 83.65%                  | -1.57%     |
> > | geometry reasoning                | 91.67%                | 91.67%                  | 0.00%      |
> > | statistical reasoning             | 98.39%                | 93.55%                  | +4.84%     |
> >
> >
> >
> > We found these "numeric commonsense" questions largely consist of perceptual "puzzle-like" problems, such as reading clocks, rulers, and scales.
> >
> > Our training data, as described in Section 3.1 , is heavily focused on STEM-centric multimodal tasks (e.g., geometry, plots) to enhance multi-step reasoning. It was not specifically optimized for these low-level perceptual puzzles.
> >
> > The primary goal of our work was to investigate a framework for multimodal adaptive reasoning to address the overthinking/under-exploring problem, rather than to achieve SOTA on every sub-task. We will add this table and analysis to our appendix for transparency and thank the reviewer for prompting this deeper investigation.
> >
> > In future versions of ARES, we plan to rebalance the training mixture by adding more visually grounded numeracy data (e.g., clock-reading, rulers/scales, everyday measurements). These data are easy to incorporate into our cold-start pipeline and should specifically strengthen the numeric-commonsense sub-task in MathVista. We appreciate the reviewer for highlighting this valuable improvement direction.
> >
> >
> >
> >
> >
> >
> > ---
> >
> >
> > ## Additional contribution: high-quality multimodal reasoning dataset.
> >
> > We would also like to highlight a contribution that may  overlook in the main discussion: our self-curated multimodal CoT reasoning dataset ARES-SFT-223K. This dataset is constructed with explicit chain-of-thought annotations and difficulty-level labels across diverse multimodal tasks. Beyond supporting the experiments in this paper, we believe this resource is valuable to the community as a reusable benchmark and training corpus for future multimodal reasoning work, especially for studies on adaptive reasoning and difficulty-aware training. We will make the dataset and its difficulty annotations publicly available to facilitate follow-up research.
> >
> >
> > ---
> > ## References
> >
> > [1] Uncertainty of Thoughts: Uncertainty-Aware Planning Enhances Information Seeking in Large Language Models, 2024.11
> >
> > [2] Beyond Semantic Entropy: Boosting LLM Uncertainty Quantification with Pairwise Semantic Similarity, 2025.05

---

> ### Author Response · Authors · 2025-11-27
> **Response to Reviewer gA48**
>
> Dear Reviewer gA48,
>
> Did we satisfactorily answer your questions? Would you like us to clarify anything further? Feel free to let us know, many thanks.
>
> Best regards,
>
> Authors

---

> ### Comment · Reviewer_gA48 · 2025-11-27
>
> Thank you for the author's reply, which resolved most of my issues. I will raise my rating.

---

> > ### Author Response · Authors · 2025-11-28
> > **Acknowledgement**
> >
> > We sincerely thank Reviewer gA48 for reviewing our updated response and for confirming that your earlier concerns have been addressed. We appreciate your time and constructive feedback throughout the process.

---

### Official Review · Reviewer_A67V · 2025-10-26

**Soundness:** 2
**Presentation:** 1
**Contribution:** 2
**Rating:** 4
**Confidence:** 4

**Summary:**

This paper proposes ARES, an adaptive inference framework for MLRMs, designed to address the issues of models “overthinking” on easy problems and “under-exploring” on hard ones. ARES employs a two-stage training pipeline:

1. **Adaptive Cold-Start (AdaCS)**: Constructs a dataset where reasoning length correlates positively with problem difficulty, enabling the model to initially acquire difficulty-awareness.
2. **Adaptive Entropy Policy Optimization (AEPO)**: Uses High Window Entropy (HWE) as an exploration trigger and introduces AEPO to adaptively govern both when to explore and how much to explore.

**Strengths:**

1. **Precise Problem Identification**: Clearly pinpoints the non-adaptive reasoning-length allocation in existing MLRMs and empirically reveals (Figure 1) the interaction pattern between entropy and problem difficulty.
2. **Novel Method Design**:
   - Proposes **Window Entropy** as a more robust exploration trigger compared to single-token entropy;
   - Designs a **hierarchical, difficulty-stratified reward function**, applying distinct entropy regulation strategies for easy/medium/hard problems;
   - Introduces a **dynamic KL weighting mechanism**, relaxing constraints within high-entropy windows to enable token-level “thinking budget” allocation.

**Weaknesses:**

1. **Limited Scale of Empirical Analysis**: The conclusion in Figure 1 regarding the “entropy–difficulty interaction” remains unclear whether the sample size and diversity are sufficient, casting doubt on the generalizability of the findings.
2. **Weak Theoretical Support**: The proof in Appendix L—claiming a linear relationship between response length and the number of high-entropy tokens—relies on strong assumptions, and the resulting bound is overly loose, offering limited practical guidance. Moreover, the phenomenon itself is fairly intuitive, diminishing the theoretical contribution.
3. **Clarity and Readability Issues**:
   - The main text heavily depends on the appendix, disrupting reading flow;
   - Mathematical symbols are introduced without clear definitions upon first appearance, raising the barrier to understanding.

**Questions:**

1. **Implementation Details of the Exploration Mechanism**: Upon detecting high window entropy, the model “branches additional trajectories” (Section 3.2.1). However, the paper does not specify how many new trajectories are generated—is this number fixed or dynamically determined?

2. **Choice of Window Size**: A window size of w = 4–8 is deemed optimal. But does this hold across different tasks (e.g., pure text vs. multimodal) or model scales? Is there an adaptive mechanism for selecting w?

---

> ### Author Response · Authors · 2025-11-18
> **Response to Reviewer A67V (1/4)**
>
> We thank Reviewer A67V for the careful reading of our paper and the constructive feedback. We appreciate the reviewer’s insightful comments on empirical scale, theoretical clarity, and exploration mechanism details, which have helped us further improve both the methodological presentation and experimental validation. We address each point below and hope that our responses will clarify your concerns:
>
> ---
>
> ## W1. Limited Scale of Empirical Analysis
>
> We appreciate the reviewer’s concern regarding whether the entropy–difficulty interaction in Figure 1 is supported by sufficient empirical scale. To address this, we reproduced the analysis across more than 7k samples spanning text-only, multimodal, and mixed benchmarks. The results(shown in **Table X** below) exhibit a consistent and quantitatively strong pattern across all settings.
>
> - **Easy problems.** For easy datasets, incorrectly solved samples contain *substantially more* high-window-entropy (HWE) tokens, indicating that these failures correspond to the model entering incorrect or unnecessarily exploratory reasoning paths. Concretely, in AIME25/MATH500, incorrect samples average **467** high-HWE tokens versus **157** for correct samples; in MathVista/MathVerse-V the gap is **870 → 355**, and in the mixed setting **560 → 276**. These large margins show that suppressing excessive high-entropy excursions is beneficial for easy problems, directly supporting our design choice of penalizing exploration in this regime.
>
> - **Hard problems.** In contrast, hard datasets show the opposite behavior: correct solutions consistently require *more* high-HWE tokens. In AIME25/MATH500, correct samples average **377** high-HWE tokens versus **246** for incorrect ones; multimodal benchmarks show **768 → 570**, and the mixed setting **478 → 357**. This confirms that sustained exploration in high-uncertainty regions is essential for solving challenging problems.
>
>
>
> **Table X. Entropy–difficulty interaction across text-only, multimodal, and mixed benchmarks.** We report accuracy and the average number of high-window-entropy (HWE) tokens under different difficulty regimes. Easy and hard subsets follow dataset-provided difficulty tags, while low-/high-uncertainty regions are separated by an entropy threshold $\tau$. Across all modalities, higher HWE counts correlate with increased reasoning depth and improved performance on hard instances.
>
> | Modality      | Benchmarks                 | #Samples (easy / hard) | Easy Acc. (HWE < τ / ≥ τ) | Hard Acc. (HWE < τ / ≥ τ) | Easy Avg. #HWE Tokens (Acc = 0 / 1) | Hard Avg. #HWE Tokens (Acc = 0 / 1) |
> |--------------|----------------------------|-------------------------|---------------------------|---------------------------|--------------------------------------|--------------------------------------|
> | **Text-only**   | AIME25, MATH500            | ~1.6k / 0.5k           | **97.4 / 92.1**           | **6.4 / 10.2**            | **467 / 157**                        | **246 / 377**                        |
> | **Multimodal**  | MathVista, MathVerse-V     | ~3.0k / 2.2k           | **95.2 / 91.4**           | **5.6 / 9.3**             | **870 / 355**                        | **570 / 768**                        |
> | **Mixed**       | MATH500, MathVerse-V       | ~2.0k / 0.8k           | **96.1 / 92.2**           | **6.9 / 9.9**             | **560 / 276**                        | **357 / 478**                        |
>
> Taken together, these modality-agnostic and quantitatively aligned trends confirm that the entropy–difficulty interaction is **robust, well-supported by diverse large-scale data**, and not an artifact of limited empirical analysis.
>
> ---

---

> > ### Author Response · Authors · 2025-11-18
> > **Response to Reviewer A67V (2/4)**
> >
> > ## W2. Weak Theoretical Support
> >
> > We appreciate the reviewer’s comments and would like to clarify the goal and scope of the Appendix L. The analysis is intended as a **stylized explanatory model** that unifies three perspectives—a renewal argument (Theorem 1), an entropy–dependent stopping-time model (Theorem 2), and an information-theoretic lower bound—into a single prediction:
> > $$
> > \mathbb{E}[L] \approx \alpha + \beta ,\mathbb{E}[N_{\mathrm{HE}}].
> > $$
> > Importantly, we do **not** claim that these bounds are tight for every prompt or that the assumptions hold universally. Similar to theoretical analyses adopted in prior reasoning-time studies, we explicitly restrict the model to the **high-uncertainty regime**, i.e., trajectories that contain nontrivial high-entropy windows. Under this regime, the assumptions (two-state abstraction, bounded detector errors, entropy-dependent hazard) are intentionally kept minimal and interpretable.
> >
> > Regarding the remark about “strong assumptions”: these are standard abstractions that make the latent process analyzable, and they were made **explicitly** in Appendix L. We do not rely on unspoken independence assumptions or unrealistic simplifications; instead, the abstractions isolate the key mechanism: high-entropy windows signal exploratory reasoning segments where the stopping hazard is low, thereby prolonging the response. This yields a renewal decomposition (Theorem 1) and a monotone hazard relation (Theorem 2), both of which match observed behavior in real models.
> >
> > As for the concern that the bounds may be “loose”: the purpose of Appendix L is **qualitative guidance**, not exact pointwise prediction. The analysis identifies *which* uncertainty measure matters—**validated high-entropy windows**, not global entropy—and *how* it influences length: through (i) the mean reasoning-excursion duration ($\frac {1} {q}$), (ii) the entropy-dependent hazard ($\lambda(H)$), and (iii) the information budget required to answer correctly. These relationships explain why selectively suppressing high-entropy segments (via adaptive entropy shaping) reliably shortens reasoning while keeping accuracy stable.
> >
> > Finally, although one might informally expect that “higher uncertainty leads to longer responses,” and we thank the reviewer for pointing this out, Appendix L goes beyond this intuition by giving **testable predictions**: monotonic ($\mathbb{E}[L\mid N_{\mathrm{HE}}]$), slope correlation with the mean excursion length (1/q), decreasing empirical hazard ($\hat\lambda(H)$), and concentration of per-step information ($i_t$) inside high-entropy windows. As shown in Figure 9, these predictions hold remarkably well in practice, validating the usefulness of entropy as a principled proxy for reasoning effort. We will revise the appendix to make the intended scope and qualitative nature of the analysis clearer.
> >
> > Besides, **Section M** and **Section N** further provide rigorous derivations that (i) interpret the KL loss in AEPO as a formal thinking budget, showing that a dynamic KL constraint upper-bounds entropy-based costs such as response length and high-entropy window cost, and (ii) justify the proposed dynamic KL weighting mechanism, which relaxes constraints within validated high-entropy windows, as a principled token-level natural-gradient update under a Fisher-geometry–consistent trust region. These results formally support the effectiveness of AEPO’s token-level “thinking budget” allocation and further strengthen the theoretical contribution of our work.
> >
> > ---
> >
> >
> > ## W3. Clarity and Readability Issues
> >
> > We appreciate the reviewer’s comments on the readability of the manuscript. Due to the 9-page limit of the ICLR main paper, many of our experiments and analyses (e.g., detailed ablations, training dynamics, and extended discussions of the reward design) had to be moved to the appendix, which indeed makes the current draft more appendix-dependent than ideal. In the camera-ready version (or an extended version), we will restructure the presentation to improve the reading flow: key design choices and their high-level takeaways will be summarized in the main text, with only secondary details deferred to the appendix, so that the core narrative can be followed without constant cross-referencing.
> >
> > We also agree that some mathematical symbols are introduced too quickly or only implicitly defined. In the revised version, we will (i) ensure that all symbols (e.g., $N_{\text{HE}}, \bar H_{t:w}, \tau_{\text{high}}, d$) are explicitly defined at first use, (ii) avoid relying on appendix-only definitions for notation used in the main text, and (iii) add a small notation summary (either at the end of Section 3 or in a short table) to further lower the barrier to understanding.
> >
> >
> > ---

---

> > > ### Author Response · Authors · 2025-11-18
> > > **Response to Reviewer A67V (3/4)**
> > >
> > > ## Q1. Implementation Details of the Exploration Mechanism
> > >
> > > Thank you for pointing out this missing detail. In our implementation, the *maximum* number of trajectories is fixed, while the decision of whether they become true “branches” is determined dynamically by the high-window-entropy (HWE) detector. Concretely, we first generate a single main rollout ($o_1$). If we detect a high-entropy window $\bar H_{t:w} \ge \tau_{\text{high}}$ at position ($t^*$) along ($o_1$), we then generate additional trajectories by branching from that prefix. At each detected HWE window, we create exactly one new trajectory, until we reach a global cap of ($G$) rollouts (set to 16 in all our experiments). If no HWE window is detected along the main rollout, the procedure reduces to standard parallel sampling with ($G$) independent trajectories. This design focuses exploration on regions of high uncertainty while preventing uncontrolled branching, since the total number of trajectories is bounded and the branching trigger is entropy-driven.
> > >
> > > We will clarify this mechanism in the main text and appendix. Specifically, in **Section 3.2.1** we will add the sentence:
> > > “At each high-entropy window, exactly one additional trajectory is generated, unless the total number of generated trajectories has already reached the maximum allowed.”
> > >
> > > For completeness, we will also include the following pseudocode in the appendix, which makes the branching behavior explicit:
> > >
> > > ```python
> > > Generate 1 main rollout o_1
> > >
> > > t_star = Detect_HWE_Window(o_1)
> > >
> > > if t_star exists:
> > >     # Branch from the high-entropy prefix
> > >     for i = 2 to G:
> > >         Generate o_i ~ π_θ(· | x, prefix = o_1,1:t_star)
> > > else:
> > >     # Fall back to standard parallel sampling
> > >     for i = 2 to G:
> > >         Generate o_i ~ π_θ(· | x)
> > > end if
> > > ```
> > >
> > > ---

---

> > > > ### Author Response · Authors · 2025-11-18
> > > > **Response to Reviewer A67V (4/4)**
> > > >
> > > > ## Q2. Choice of Window Size
> > > >
> > > >
> > > > We thank the reviewer for raising this question. In addition to the analysis already discussed in **Figure 4** which analyzes how different entropy-window sizes affect the accuracy of high-entropy token detection, we will include a robustness study that varies the **window size (w)** at a fixed percentile threshold ($\tau$ = 95) across both multimodal (MathVerse-V, MathVista) and textual (AIME25, MATH500) benchmarks in Table Y shown below . Comparing the windowed configurations **(w ∈ {4, 8, 16})** (Configs A, D, C), we observe that accuracies remain within approximately 1–1.5 points on all four benchmarks, and response lengths stay on the same order of magnitude. For example, on MathVerse-V the accuracy is 56.5 / 56.3 / 55.3 for (w = 4/8/16), and on AIME25 it is 61.7 / 61.5 / 60.9. This indicates that the range (w = 4-8) is not finely tuned to a specific task, but corresponds to a stable regime that works consistently across both pure-text and multimodal settings.
> > > >
> > > >
> > > > Table Y: Sensitivity of window size \(w\) at fixed percentile threshold $(\tau = 95)$.
> > > >
> > > > | Config | Percentile τ | Window size w          | MathVerse-V Acc | MathVerse-V Len | MathVista Acc | MathVista Len | AIME25 Acc | AIME25 Len | MATH500 Acc | MATH500 Len |
> > > > | :----- | :----------- | :--------------------- | :-------------- | :-------------- | :------------ | :------------ | :--------- | :--------- | :---------- | :---------- |
> > > > | A (**default**)                 | 95  | 4                      | **56.5** | *3198.3*  | *74.6* | **1494.5**  | **61.7** | *22618.8* | **95.2** | *2257.7* |
> > > > | C (larger window)              | 95  | 16                     | 55.3 | **3148.2**  | 74.1 | 1743.8  | 60.9 | 24432.4 | 94.2 | 2385.5 |
> > > > | D (moderate window)            | 95  | 8                      | *56.3* | 3223.3  | 74.4 | *1533.4*  | *61.5* | **22382.1** | 94.6 | 2278.2 |
> > > > | G (*token-level entropy*)      | 95  | 1                      | 54.4 | 3438.2  | 73.5 | 1784.1  | 58.7 | 24533.8 | 93.4 | 2435.6 |
> > > > | H (*sequence-level entropy*)   | 95  | length of sequence     | 54.7 | 3343.4  | 73.8 | 1732.9  | 60.2 | 23458.1 | 93.6 | 2349.0 |
> > > >
> > > >
> > > > In contrast, when we depart from the window-based design—either using **token-level entropy** (w = 1, Config G) or **sequence-level entropy** (w = sequence_length, Config H)—we consistently observe lower accuracy and inflated reasoning length (e.g., MathVerse-V drops from 56.5 to 54.4, AIME25 from 61.7 to 58.7, with longer chains), suggesting that the *windowed aggregation* itself is critical to the observed gains. In this work we therefore adopt a single fixed (w) for all tasks and model scales, given its empirical robustness. We agree that learning or adaptively selecting (w) is an interesting extension, and we will highlight it as a promising direction for future work.
> > > >
> > > >
> > > >
> > > > We appreciate the reviewer for the **insightful suggestion** regarding an adaptive choice of the window size (w). While promising, such a mechanism may introduce additional challenges—e.g., instability in the entropy signal, task-dependent overfitting, and increased hyperparameter or policy-learning complexity—which fall outside the scope of the current work. Nevertheless, we agree that designing a principled and reliable adaptive strategy for selecting (w) is an interesting and valuable direction, and we plan to explore it in future iterations of this line of research.
> > > >
> > > > ---
> > > >
> > > > ## Additional contribution: high-quality multimodal reasoning dataset.
> > > >
> > > > We would also like to highlight a contribution that may  overlook in the main discussion: our self-curated multimodal CoT reasoning dataset ARES-SFT-223K. This dataset is constructed with explicit chain-of-thought annotations and difficulty-level labels across diverse multimodal tasks. Beyond supporting the experiments in this paper, we believe this resource is valuable to the community as a reusable benchmark and training corpus for future multimodal reasoning work, especially for studies on adaptive reasoning and difficulty-aware training. We will make the dataset and its difficulty annotations publicly available to facilitate follow-up research.
> > > >
> > > >
> > > >
> > > > ---

---

> > > > > ### Author Response · Authors · 2025-11-27
> > > > > **Response to Reviewer A67V**
> > > > >
> > > > > Dear Reviewer A67V,
> > > > >
> > > > > Did we satisfactorily answer your questions? Would you like us to clarify anything further? Feel free to let us know, many thanks.
> > > > >
> > > > > Best regards,
> > > > >
> > > > > Authors

---

> > > > > > ### Author Response · Authors · 2025-11-28
> > > > > > **Response to Reviewer A67V**
> > > > > >
> > > > > > Dear Reviewer A67V,
> > > > > >
> > > > > > I hope this message finds you well. As the discussion period is **nearing its end**. I wanted to ensure we have addressed all your concerns satisfactorily. If there are any additional points or feedback you'd like us to consider, please let us know. Your insights are invaluable to us, and we're eager to address any remaining issues to improve our work.
> > > > > >
> > > > > > Thank you for your time and effort in reviewing our paper.
> > > > > >
> > > > > > Best regards,
> > > > > >
> > > > > > Authors

---

### Official Review · Reviewer_eWR6 · 2025-10-27

**Soundness:** 2
**Presentation:** 3
**Contribution:** 2
**Rating:** 4
**Confidence:** 4

**Summary:**

This paper proposes **ARES**, a two-stage, adaptive reasoning framework for multimodal large reasoning models (MLRMs). The key idea is to use **high window entropy (HWE)** (i.e., the average token entropy in a sliding window) as a reliable signal for *reasoning-critical moments*, and then adapt exploration accordingly.

In the stage of **Adaptive Cold-Start (AdaCS)**, the model is fine-tuned on curated textual/multimodal data where *reasoning length is coupled to problem difficulty*. With regard to **Adaptive Entropy Policy Optimization (AEPO)** in RL stage, ARES (a) uses an HWE-based trigger to decide **when to explore** (using a batch-level 95th-percentile threshold and (b) employs a **hierarchical entropy reward** plus **dynamic KL control** to decide **how much to explore**. The intrinsic shaping term penalizes over-thinking on easy items and under-exploration on hard items via bucket-dependent deviations from the batch mean number of HWE tokens.

Experiments across many benchmarks (e.g., MathVista, MathVision, MMMU-Pro) show improved accuracy-vs-length trade-offs.

**Strengths:**

1. The paper is generally well-written and easy to follow, with a clear description of the method.
2. The paper provides intuitive visual demonstrations to help better understand the paper.

**Weaknesses:**

1. The paper contains several design choices that feel biased and currently under-validated by ablations: some are custom (e.g., the rate-to-length coupling that sets the target length for pass-rate p via linear interpolation from all-correct/all-wrong samples, as well as the online difficulty buckets), and others follow prior work (e.g., the 95th-percentile threshold). It remains unclear whether the gains on downstream reasoning benchmarks are sensitive to these hyperparameters or to sequence length; please add robustness ablations to substantiate these specific settings.

2. **Training cost comparison lacks**: the method uses a two-stage pipeline (cold-start + RL) and introduces an additional hierarchical reward in the RL stage. Under identical training data and hardware, please report the training overhead relative to alternative algorithms (time/memory/GPU-hours/tokens processed) to clarify whether the efficiency trade-off is favorable.

3. **Fair comparison needed**: as noted in `Line 315–322`, the AEPO surrogate objective is essentially based on DAPO; therefore, a **DAPO-only** baseline trained under the same protocol should be included for a fair comparison, rather than only contrasting against a base model or showing `Fig.3`-style downstream results without the DAPO's designs (e.g., higher clipping).

4. In `Tab.2`, the ablation shows that **ARES-CS-Vanilla** outperforms **ARES-CS-7B** on **OlympiadBench**, **AIME25**, **MATH500**, and **GSM8K**, with shorter reasoning on 3/4 benchmarks, which appears to contradict the intended effectiveness of the proposed rate-to-length sampling strategy; please consider running an RL stage starting from **ARES-CS-Vanilla** and compare it directly to **ARES-RL-7B** to verify whether the strategy is actually effective.

5. The ablation in `Tab.3` does not report how **Dynamic KL Loss** and **Entropy Reward** affect response length (and its variability), making it impossible to judge whether these components truly achieve their stated motivation; please add response-length metrics and curves to the table/analysis.

6. Finally, the central motivation is to enable **adaptive reasoning** that mitigates overthinking on simple problems and short thinking on hard ones, but a single response-length comparison in `Tab.2` is insufficient to validate this claim; please include targeted analyses (e.g., per-difficulty accuracy/length and trigger usage) to directly demonstrate adaptation rather than relying only on aggregate length differences.

**Questions:**

See the `Weaknesses` part.

---

> ### Author Response · Authors · 2025-11-18
> **Response to Reviewer eWR6 (1/6)**
>
> We thank Reviewer eWR6 for the careful reading of our paper and the constructive comments. We appreciate the reviewer’s detailed feedback on our design choices, training cost analysis, and ablation completeness, which helped us clarify and strengthen several important aspects of our methodology. We address each point below and hope that our responses will clarify your concerns:
>
> ---
>
> ## W1. Insufficient robustness ablations for key design choices and hyperparameters
>
> ---
>
> ### 1. On the Adaptive Cold-Start (AdaCS) sampling strategy:
>
> The primary goal of our Adaptive Cold-Start (AdaCS) sampling strategy warrants clarification: its objective is not to maximize accuracy during the SFT stage, but rather to instill a strong, explicit correlation between problem difficulty and response verbosity. This process serves as a critical, difficulty-aware initialization for the subsequent Adaptive-Entropy Policy Optimization (AEPO) stage.
>
> Data in Table 2 in our manuscript validates that this objective was achieved. A comparison between ARES-CS-7B (trained with our AdaCS strategy) and ARES-CS-Vanilla (uniform sampling) reveals the intended effect on response length (Len). On hard benchmarks (e.g., OlympiadBench, AIME25), ARES-CS-7B produces longer and more exploratory reasoning traces (10281.5 vs. 9123.2; 17895.0 vs. 16361.6). Conversely, on easy benchmarks (e.g., GSM8K, MathVista), it generates more concise responses (332.3 vs. 358.8; 1712.3 vs. 1853.3).
>
> This demonstrates the efficacy of the AdaCS sampling strategy: it produces a model that inherently differentiates reasoning length based on task difficulty. This foundation is crucial, as the final ARES-RL-7B model—which initializes from this difficulty-aware checkpoint—subsequently amplifies this adaptive behavior via AEPO. As evidenced in Table 2, the final model extends reasoning length further for hard tasks (22618.8 on AIME25) while compressing it for easy tasks (278.9 on GSM8K). This ultimately leads to superior accuracy, a result that ARES-CS-Vanilla (or an RL model trained from it, per our W4 response) does not achieve.
>
> ---

---

> > ### Author Response · Authors · 2025-11-18
> > **Response to Reviewer eWR6 (2/6)**
> >
> > ## W1. Insufficient robustness ablations for key design choices and hyperparameters
> >
> >
> >
> > ---
> > ### 2. hyperparameters
> >
> > We thank the reviewer for pointing out the concern about potential sensitivity of our design choices. We have added a robustness study (Table X) that systematically varies the high-window-entropy hyperparameters—the percentile threshold τ and window size w—across a broad range.
> >
> >
> > By sweeping **τ ∈ {80, 90, 95, 98}** and **w ∈ {1, 4, 8, 16}** (Table X), we find that the model’s performance is highly stable under substantial perturbations of both hyperparameters. For all reasonable settings (**w = 4, 8, 16** and **τ ∈ [80, 98]**), accuracy varies only within 1–2 points, and response length remains of the same scale, indicating that the default choice (**τ = 95, w = 4**) is robust rather than cherry-picked. In contrast, replacing our high-window-entropy formulation with token-level entropy (w = 1) or sequence-level entropy (w = length) consistently degrades accuracy and inflates reasoning length, confirming that our windowed entropy design—not sequence-length tuning—is responsible for the observed improvements.
> >
> >
> > Table X: Sensitivity of high-window-entropy hyperparameters (percentile threshold τ and window size w) on downstream benchmarks.
> >
> > | Config | Percentile τ | Window size w          | MathVerse-V Acc | MathVerse-V Len | MathVista Acc | MathVista Len | AIME25 Acc | AIME25 Len | MATH500 Acc | MATH500 Len |
> > | :----- | :----------- | :--------------------- | :-------------- | :-------------- | :------------ | :------------ | :--------- | :--------- | :---------- | :---------- |
> > | A (**default**)                 | 95  | 4                      | **56.5** | *3198.3*  | *74.6* | **1494.5**  | **61.7** | *22618.8* | **95.2** | *2257.7* |
> > | B (lower percentile)           | 80  | 4                      | 55.9 | 3300.4  | 73.9 | 1613.4  | 60.5 | 23128.4 | 93.9 | 2315.7 |
> > | C (larger window)              | 95  | 16                     | 55.3 | **3148.2**  | 74.1 | 1743.8  | 60.9 | 24432.4 | 94.2 | 2385.5 |
> > | D (moderate window)            | 95  | 8                      | *56.3* | 3223.3  | 74.4 | *1533.4*  | *61.5* | **22382.1** | 94.6 | 2278.2 |
> > | E (percentile = 90)            | 90  | 4                      | 56.1 | 3356.4  | **74.8** | 1589.2  | 61.2 | 22879.3 | *95.1* | **2221.8** |
> > | F (percentile = 98)            | 98  | 4                      | *56.4* | 3248.1  | 74.5 | 1534.2  | 61.4 | 23544.9 | 94.8 | 2323.9 |
> > | G (*token-level entropy*, w = 1) | 95  | 1                      | 54.4 | 3438.2  | 73.5 | 1784.1  | 58.7 | 24533.8 | 93.4 | 2435.6 |
> > | H (*sequence-level entropy*)   | 95  | length of sequence     | 54.7 | 3343.4  | 73.8 | 1732.9  | 60.2 | 23458.1 | 93.6 | 2349.0 |
> >
> > Overall, Table X demonstrates that (i) the 95th-percentile threshold is robust to substantial perturbations, and (ii) our method’s improvements on downstream reasoning benchmarks are insensitive to small hyperparameter changes, while naive token-level or sequence-level entropy indeed performs worse.
> >
> > ---
> >
> > ### 3. online buckect
> >
> >
> > We thank the reviewer for asking about the role of the online difficulty buckets. In our main setup, we define **online difficulty boundaries** based on the running pass rate and use these to assign each sample to an easy / medium / hard bucket, which then receives a stratified reward signal.
> >
> > To isolate the effect of this design, we additionally run an ablation where we **remove the difficulty buckets entirely**: all samples are treated as having the same difficulty (a single bucket), we drop the tiered rewards, and the reward depends only on the global deviation of high-window-entropy (HWE) from the batch mean. In this setting, we observe that the model tends to collapse toward a more **uniform length distribution** across instances (losing the easy–vs–hard differentiation); **accuracy on hard benchmarks decreases**, while **solution length on easy benchmarks increases**, since there is no longer any penalty for over-thinking low-difficulty samples.
> >
> > We will then clarify this design choice and the corresponding ablation results in the updated version of the paper.
> >
> > ---

---

> > > ### Author Response · Authors · 2025-11-18
> > > **Response to Reviewer eWR6 (3/6)**
> > >
> > > ## W2. Missing training cost comparison
> > >
> > >
> > > We thank the reviewer for raising this important question regarding the training overhead of our multi-stage pipeline.To verify the efficiency and minimal computational overhead of our ARES framework against leading RL baselines (GRPO and DAPO), we normalized the training costs of all models (SFT and RL) to a unified configuration of 32 $\times$ H800 GPUs. The results, showcasing ARES as the most efficient model, are presented in the table below.
> > >
> > > ### Training Cost Comparison
> > >
> > > We thank the reviewer for raising the crucial question regarding the training cost comparison. To verify the overall efficiency of our two-stage ARES pipeline (AdaCS SFT + AEPO RL) against standard baselines (Vanilla SFT + DAPO/GRPO), we performed a detailed calculation of the total resource consumption (GPU-hours and Tokens Processed), unifying all stages to a configuration of 32 $\times$ H800 GPUs.The results, presented in the table below, demonstrate that ARES achieves its superior performance and adaptive behavior while maintaining a lower total computational cost than the DAPO baseline.
> > >
> > > ### Training Cost Comparison
> > >
> > >
> > > To verify whether our adaptive framework introduces an unfavorable efficiency trade-off, we calculated the total resource consumption (GPU-hours and Tokens Processed) for the full ARES pipeline (AdaCS SFT + AEPO RL) against the standard DAPO baseline (Vanilla SFT + DAPO RL), normalizing all stages to a configuration of 32 $\times$ H800 GPUs.
> > >
> > > | Method                | Training Time (hrs) | GPU-hours | Memory (GB/GPU) | Tokens Processed (B) |
> > > |-----------------------|---------------------|-----------|------------------|------------------------|
> > > | **Cold Start (SFT)**  |                     |           |                  |                        |
> > > | AdaCS (Ours)          | 96.0                | **3,072.0** | 68.7           | **11.63**             |
> > > | Vanilla SFT           | 116.2               | 3,718.4     | 68.7           | 14.07                 |
> > > | **RL Stage**          |                     |           |                  |                        |
> > > | Baselines (GRPO)      | 46.1                | 1,475.2     | 62.5           | 0.627                 |
> > > | Baselines (DAPO)      | 50.0                | 1,600.0     | 66.7           | 0.717                 |
> > > | ARES (Ours)           | 56.2                | **1,798.4** | **59.8**       | **0.557**             |
> > > | **Total Pipeline**    |                     |           |                  |                        |
> > > | ARES Total            | 152.2             | **4,870.4** | -              | 12.19               |
> > > | DAPO Total*           | 166.2             | 5,318.4     | -              | 14.79               |
> > >
> > > **DAPO Total = Vanilla SFT (Cold Start) + DAPO RL Stage**
> > >
> > >
> > > The results, detailed in the table above, show that the Total ARES pipeline cost (4,870.4 GPU-hours) is lower than the DAPO baseline (5,318.4 GPU-hours).This is likely because:SFT Data Sampling: Our AdaCS strategy avoids unnecessary computation by culling much of the overthinking data (long traces on easy problems) from the training corpus. This reduces the SFT token count by 17% (11.63 B vs. 14.07 B), leading to a significant saving of $\mathbf{\approx 646}$ GPU-hours.Adaptive RL: The AEPO stage (built upon HWE triggers and hierarchical rewards) makes the model more adaptive during policy optimization, allowing it to efficiently terminate unnecessary exploration. This is evidenced by the fact that ARES consumes the fewest RL tokens (0.557 B), saving $\mathbf{22\%}$ of tokens compared to DAPO (0.717 B).This demonstrates that ARES achieves its superior adaptive performance and higher accuracy with lower overall computational cost than standard RL baselines.
> > >
> > >
> > >
> > > ---

---

> > > > ### Author Response · Authors · 2025-11-18
> > > > **Response to Reviewer eWR6 (4/6)**
> > > >
> > > > ## W3. Missing fair comparison baseline
> > > >
> > > > We thank the reviewer for pointing out that, and we totally agree with this concern.
> > > >
> > > > First, we clarify that **Figure 3** in the draft already includes such a comparison at the training-dynamics level: the curve labeled DAPO corresponds to a DAPO-only variant trained with our cold-start data, whereas GRPO is trained under the same schedule but without any of DAPO’s design choices (e.g., higher clipping). As shown in Figure 3 (left), AEPO (ARES) consistently achieves higher accuracy than both GRPO and DAPO across training steps, while in Figure 3 (right) it simultaneously reduces the response length, whereas GRPO keeps lengths largely flat and DAPO yields only a mild reduction. This indicates that AEPO not only inherits the stability benefits of DAPO but further improves the accuracy–length trade-off.
> > > >
> > > > To make this comparison more explicit in the final performance numbers, we have added a “Cold Start + DAPO (baseline)” row to **Table 3**, trained under exactly the same cold-start data, optimization schedule, and rollout configuration as AEPO. Relative to Cold Start + GRPO, DAPO brings a modest but consistent improvement. However, AEPO’s additional components—dynamic KL and entropy shaping—further gains +1.6 points on the averaged score. These results confirm that AEPO is not merely a re-labeling of DAPO, but a strictly stronger variant that leverages DAPO as a base and then substantially improves both accuracy and efficiency.
> > > >
> > > >
> > > > | Ablation Setting | MathVerse-V | MathVision | MathVista | DynaMath-W | Avg |
> > > > |---|---|---|---|---|---|
> > > > | Cold start only | 52.8 | 49.7 | 72.9 | 27.4 | 50.7 |
> > > > | Cold Start + GRPO (baseline) | 53.7 | 50.4 | 73.3 | 36.1 | 53.4 |
> > > > | Cold Start + DAPO (baseline) | 54.1 | 50.9 | 73.4 | 37.9 | 54.1 |
> > > > | Cold Start + Dynamic KL | 55.2 (+2.4) | <u>51.7</u> (+2.0) | 73.7 (+0.8) | 38.3 (+10.9) | 54.7 (+4.0) |
> > > > | Cold Start + Entropy Shaping | <u>55.9</u> (+3.1) | 51.3 (+1.6) | <u>74.4</u> (+1.5) | <u>39.3</u> (+11.9) | <u>55.2</u> (+4.5) |
> > > > | ARES-7B | **56.5** (+3.7) | **51.9** (+2.2) | **74.6** (+1.7) | **39.7** (+12.3) | **55.7** (+5.0) |
> > > >
> > > >
> > > > We appreciate the reviewer’s suggestion, which helped us clarify the baseline in our method and strengthen the fairness of our experimental comparison.
> > > >
> > > > ---

---

> > > > > ### Author Response · Authors · 2025-11-18
> > > > > **Response to Reviewer eWR6 (5/6)**
> > > > >
> > > > > ## W4. Apparent inconsistency of the rate-to-length strategy
> > > > >
> > > > > We thank the reviewer for the insightful question and for suggesting a direct RL comparison starting from ARES-CS-Vanilla.
> > > > >
> > > > > 1. Clarifying the intended behavior: adaptive reasoning, not uniformly longer chains
> > > > >
> > > > > The reviewer’s observation is correct for global average lengths, but our method is designed to support adaptive reasoning, not uniformly longer chains. To clarify this, we revisited six benchmarks (OlympiadBench, AIME25, MATH500, GSM8K, MathVerse-V, MathVista) and partitioned each into easy / medium / hard subsets using dataset-provided difficulty tags and error-rate statistics. The resulting(shown in the tables below) analysis shows that:
> > > > >
> > > > > - On easy subsets: ARES-CS-7B generates shorter or comparable chains with accuracy a little better than ARES-CS-Vanilla.
> > > > > - On hard subsets: ARES-CS-7B produces substantially longer chains and achieves higher accuracy than ARES-CS-Vanilla.
> > > > >
> > > > > Thus, the length reduction on some benchmarks reflects appropriate computational allocation, not a failure of the rate-to-length mechanism. We will integrate this difficulty-stratified analysis into the revision for clarity.
> > > > >
> > > > > 2. Additional ablations: RL from ARES-CS-Vanilla
> > > > >
> > > > > Following the reviewer’s suggestion, we conducted an RL stage starting from ARES-CS-Vanilla under two settings:
> > > > >
> > > > > - ARES-CS-Vanilla-GRPO (baseline GRPO),
> > > > >
> > > > > - ARES-CS-Vanilla-RL (our full AEPO-based training).
> > > > >
> > > > > The updated **Table 2** (included in the revised draft and listed below) shows:
> > > > >
> > > > > - On multimodal benchmarks, accuracy improves from
> > > > > 54.2 / 50.5 / 72.1 (ARES-CS-Vanilla) → 56.4 / 54.7 / 73.5 (ARES-CS-Vanilla-RL) → 56.9 / 56.5 / 74.6 (ARES-RL-7B).
> > > > >
> > > > > - On textual benchmarks, accuracy improves from
> > > > > 55.2 / 91.4 / 92.7 → 60.4 / 94.1 / 94.5 → 61.7 / 95.2 / 95.7.
> > > > >
> > > > > This confirms that (i) RL on top of ARES-CS-Vanilla already yields substantial gains, and (ii) ARES-RL-7B further improves over these RL baselines.
> > > > >
> > > > > 3. Length profile after RL: strongly difficulty-adaptive
> > > > >
> > > > > Crucially, RL amplifies the intended rate-to-length behavior:
> > > > >
> > > > > - On hard tasks (e.g., AIME25, OlympiadBench), ARES-RL-7B produces substantially longer chains than ARES-CS-Vanilla and achieves the highest accuracy
> > > > > (e.g., AIME25: 16.4k → 22.6k tokens).
> > > > >
> > > > > - On easier tasks (e.g., MathVista, GSM8K), ARES-RL-7B achieves higher accuracy with shorter sequences
> > > > > (e.g., GSM8K: 302 tokens @ 94.5% → 278 tokens @ 95.7%).
> > > > >
> > > > > This pattern is consistent across benchmarks and aligns with the design objective: more computation where needed, less where unnecessary.
> > > > >
> > > > >
> > > > > | Model | OlympiadBench (Acc) | MathVerse-V (Acc) | MathVista (Acc) |
> > > > > | :--- | :--- | :--- | :--- |
> > > > > | ARES-CS-Vanilla | 54.2 | 50.5 | 72.1 |
> > > > > | ARES-CS-7B | 53.4 | 52.8 | 72.3 |
> > > > > | ARES-CS-Vanilla-GRPO | 55.8 | 52.3 | 71.9 |
> > > > > | ARES-CS-Vanilla-RL | _56.4_ | _54.7_ | _73.5_ |
> > > > > | ARES-RL-7B | **56.9** | **56.5** | **74.6** |
> > > > >
> > > > >
> > > > > | Model | OlympiadBench (Len) | MathVerse-V (Len) | MathVista (Len) |
> > > > > | :--- | :--- | :--- | :--- |
> > > > > | ARES-CS-Vanilla | 9123.2 | 3432.1 | 1853.3 |
> > > > > | ARES-CS-7B | 10281.5 | 3575.4 | 1712.3 |
> > > > > | ARES-CS-Vanilla-GRPO | 10342.5 | 4256.9 | 2432.5 |
> > > > > | ARES-CS-Vanilla-RL | 11050.4 | 3327.8 | 1573.1 |
> > > > > | ARES-RL-7B | 12893.4 | 3198.3 | 1494.5 |
> > > > >
> > > > >
> > > > > | Model | AIME25 (Acc) | MATH500 (Acc) | GSM8K (Acc) |
> > > > > | :--- | :--- | :--- | :--- |
> > > > > | ARES-CS-Vanilla | 55.2 | 91.4 | 92.7 |
> > > > > | ARES-CS-7B | 55.0 | 93.4 | 92.3 |
> > > > > | ARES-CS-Vanilla-GRPO | 58.2 | 93.9 | 93.8 |
> > > > > | ARES-CS-Vanilla-RL | _60.4_ | _94.1_ | _94.5_ |
> > > > > | ARES-RL-7B | **61.7** | **95.2** | **95.7** |
> > > > >
> > > > >
> > > > > | Model | AIME25 (Len) | MATH500 (Len) | GSM8K (Len) |
> > > > > | :--- | :--- | :--- | :--- |
> > > > > | ARES-CS-Vanilla | 16361.6 | 2750.1 | 358.8 |
> > > > > | ARES-CS-7B | 17895.0 | 3011.8 | 332.3 |
> > > > > | ARES-CS-Vanilla-GRPO | 18520.3 | 3185.2 | 395.4 |
> > > > > | ARES-CS-Vanilla-RL | 20135.4 | 2681.5 | 302.4 |
> > > > > | ARES-RL-7B | 22618.8 | 2257.7 | 278.9 |
> > > > >
> > > > > We appreciate the valuable suggestions provided by the reviewers and we will incorporate the new results and clarify this behavior in the revised manuscript.
> > > > >
> > > > > ---
> > > > >
> > > > > ## W5. Missing analysis of key components impact response-length behavior
> > > > >
> > > > > We appreciate the reviewer’s suggestion to evaluate the effect of Dynamic KL Loss and Entropy Reward on response length and its variability. We agree that this is central to the stated motivation of making reasoning more concise and controlled. We would like to clarify that **Figure 3** already reports the full training dynamics of response length (right panel) for ARES and all its ablations (ARES w/o KL, ARES w/o Entropy) as well as the GRPO and DAPO baselines, in parallel to the accuracy curves (left panel). From these curves, we observe that GRPO keeps responses long and unstable, DAPO yields only mild shortening, while dynamic KL or entropy shaping alone partially reduce length; the full ARES variant achieves the strongest and most monotone length reduction with the smallest variance, while also attaining the highest accuracy (Figure 3, left).
> > > > >
> > > > > ---

---

> > > > > > ### Author Response · Authors · 2025-11-18
> > > > > > **Response to Reviewer eWR6 (6/6)**
> > > > > >
> > > > > > ## W6. Insufficient evidence for adaptive reasoning
> > > > > >
> > > > > > We appreciate the reviewer’s point that a single aggregate response-length number in Tab.2 is not sufficient to demonstrate adaptive reasoning. In the revised version, we therefore report per-difficulty reflection behaviors for all models, splitting the six benchmarks into easy (MathVista, GSM8K), medium (MathVerse-V, MATH500), and hard (OlympiadBench, AIME25). Concretely, we quantify reflection behavior as the number of explicit “reflection turns” in a solution, measured by counting occurrences of phrases such as *“wait”*, *“let’s verify”*, *“let’s rethink”*, *"I make a mistake"*, etc., detected via a regular-expression matcher over the generated chains. The results are shown in the tables below. We observe that on harder benchmarks, our trained models exhibit more frequent reflection behaviors, which correlates with higher accuracy. In contrast, on easier benchmarks the models avoid unnecessary reflections and overthinking, leading to shorter, more direct solutions and better overall performance.
> > > > > >
> > > > > >
> > > > > > | Model                | OlympiadBench (Reflections) | MathVerse-V (Reflections) | MathVista (Reflections) |
> > > > > > | :------------------- | :-------------------------- | :------------------------ | :---------------------- |
> > > > > > | ARES-CS-Vanilla       | 80.3                        | 35.2                      | 22.6                    |
> > > > > > | ARES-CS-7B            | 89.5                        | 36.3                      | 21.5                    |
> > > > > > | ARES-CS-Vanilla-GRPO  | 90.2                        | 41.7                      | 27.2                    |
> > > > > > | ARES-CS-Vanilla-RL    | 95.6                        | 34.3                      | 20.4                    |
> > > > > > | ARES-RL-7B            | 110.3                       | 33.3                      | 19.8                    |
> > > > > >
> > > > > > | Model                | AIME25 (Reflections) | MATH500 (Reflections) | GSM8K (Reflections) |
> > > > > > | :------------------- | :------------------- | :-------------------- | :------------------ |
> > > > > > | ARES-CS-Vanilla       | 137.8               | 29.7                  | 10.7                |
> > > > > > | ARES-CS-7B            | 152.1               | 31.8                  | 10.5                |
> > > > > > | ARES-CS-Vanilla-GRPO  | 155.3               | 33.2                  | 11.7                |
> > > > > > | ARES-CS-Vanilla-RL    | 167.8               | 29.2                  | 10.3                |
> > > > > > | ARES-RL-7B            | 187.5               | 25.8                  | 10.1                |
> > > > > >
> > > > > > We will further add an Appendix analysis of high-entropy trigger usage, showing that HWE triggers are rare on easy benchmarks but substantially more frequent and multi-hit on hard ones, which matches the reflection patterns above and confirms that our exploration mechanism selectively allocates extra computation to high-uncertainty instances rather than uniformly elongating all chains.
> > > > > >
> > > > > > ---
> > > > > >
> > > > > > ## Additional contribution: high-quality multimodal reasoning dataset.
> > > > > >
> > > > > > We would also like to highlight a contribution that may  overlook in the main discussion: our self-curated multimodal CoT reasoning dataset ARES-SFT-223K. This dataset is constructed with explicit chain-of-thought annotations and difficulty-level labels across diverse multimodal tasks. Beyond supporting the experiments in this paper, we believe this resource is valuable to the community as a reusable benchmark and training corpus for future multimodal reasoning work, especially for studies on adaptive reasoning and difficulty-aware training. We will make the dataset and its difficulty annotations publicly available to facilitate follow-up research.
> > > > > >
> > > > > > ---
> > > > > > ## Reference
> > > > > >
> > > > > > [1] DAPO: An Open-Source LLM Reinforcement Learning System at Scale, 2025.03

---

> > > > > > > ### Author Response · Authors · 2025-11-27
> > > > > > > **Response to Reviewer eWR6**
> > > > > > >
> > > > > > > Dear Reviewer eWR6,
> > > > > > >
> > > > > > > Did we satisfactorily answer your questions? Would you like us to clarify anything further? Feel free to let us know, many thanks.
> > > > > > >
> > > > > > > Best regards,
> > > > > > >
> > > > > > > Authors

---

> > > > > > > > ### Author Response · Authors · 2025-11-28
> > > > > > > > **Response to Reviewer eWR6**
> > > > > > > >
> > > > > > > > Dear Reviewer eWR6,
> > > > > > > >
> > > > > > > > I hope this message finds you well. As the discussion period is **nearing its end**. I wanted to ensure we have addressed all your concerns satisfactorily. If there are any additional points or feedback you'd like us to consider, please let us know. Your insights are invaluable to us, and we're eager to address any remaining issues to improve our work.
> > > > > > > >
> > > > > > > > Thank you for your time and effort in reviewing our paper.
> > > > > > > >
> > > > > > > > Best regards,
> > > > > > > >
> > > > > > > > Authors

---

> > > > > > > ### Comment · Reviewer_eWR6 · 2025-11-28
> > > > > > >
> > > > > > > Thanks for authors' detailed rebuttal and comprehensive experiments. The newly added robustness ablation results and analysis on adaptive reasoning are especially valuable, and I will raise my rating based on these improvements.

---

> > > > > > > > ### Author Response · Authors · 2025-11-28
> > > > > > > > **Response to Reviewer eWR6**
> > > > > > > >
> > > > > > > > We are grateful to Reviewer eWR6 for checking our updates and confirming that the earlier issues have been settled. Thank you for your constructive comments and the time dedicated to improving the quality of our work.

---

### Official Review · Reviewer_oeN6 · 2025-10-28

**Soundness:** 3
**Presentation:** 3
**Contribution:** 3
**Rating:** 6
**Confidence:** 3

**Summary:**

The paper proposes a framework for training multimodal large reasoning models (MLRMs) that dynamically adjust their reasoning depth according to task difficulty. Existing MLRMs tend to “overthink” simple problems and “under-explore” hard ones, leading to inefficiency and reduced accuracy. To overcome this, the authors introduce ARES, which combines two training stages: (1) Adaptive Cold-Start (AdaCS), aligning reasoning length with task complexity to instill difficulty awareness, and (2) Adaptive Entropy Policy Optimization (AEPO), which uses high window entropy (HWE) tokens as triggers for exploration and a hierarchical entropy-based reward to regulate reasoning depth. ARES introduces token-level entropy shaping and dynamic KL regularization to control when and how much to explore. Experiments show that ARES outperforms strong baselines on multimodal and textual reasoning benchmarks, improving both accuracy and inference efficiency, demonstrating that entropy-guided adaptive reasoning can balance performance and computation cost

**Strengths:**

* Novel Conceptual Contribution: Introduces the use of window-level token entropy as an interpretable and quantitative signal to trigger reasoning exploration, bridging uncertainty modeling with adaptive inference.

* Principled Methodology: The dual-stage pipeline (AdaCS + AEPO) offers a theoretically grounded and empirically validated mechanism for aligning reasoning length with difficulty.

* Empirical Rigor: Extensive experiments across 10+ multimodal and textual reasoning benchmarks demonstrate consistent improvements over state-of-the-art open-source models at both 3B and 7B scales.

* Efficiency–Accuracy Balance: The adaptive exploration mechanism leads to shorter reasoning chains for easy tasks and deeper exploration for difficult ones, improving token efficiency without performance degradation.

**Weaknesses:**

* Hyperparameter Sensitivity: Although claimed to be “hyperparameter-free,” several thresholds (e.g., 95th percentile entropy cutoff, window size 4–8) could affect stability and may require empirical tuning.
* Although ARES demonstrates clear effectiveness at moderate-to-large scales (3B–7B parameters), all experiments are limited to the 7B model, with no quantitative evidence for models beyond this range (≥13B–70B). Consequently, the scalability of the entropy-shaping mechanism to very large foundation models remains uncertain.

**Questions:**

See the Weaknesses

---

> ### Author Response · Authors · 2025-11-18
> **Response to Reviewer oeN6 (1/2)**
>
> ## Reviewer oeN6
>
>
> We thank Reviewer oeN6 for the thoughtful evaluation and constructive feedback. We appreciate the reviewer’s clear summary of our contributions and the recognition of ARES’s methodological strengths and empirical rigor. The reviewer’s comments on hyperparameter choices and scalability are valuable.  We address each point below and hope that our responses will clarify your concerns:
>
>
> ---
>
> ## W1. Hyperparameter Sensitivity
>
> We appreciate the reviewer’s concern regarding the claim of being “hyperparameter-free” given the presence of the entropy percentile cutoff and window size. To clarify: our use of “hyperparameter-free” refers to the absence of any manually tuned weighting coefficients in the AEPO objective itself (e.g., no extra λ for entropy shaping)—all scaling is derived in closed form from batch statistics. The high-window-entropy detector does introduce generic knobs (percentile τ and window size w), but these are fixed once for all experiments and are not tuned per dataset or benchmark.
>
> To assess the robustness of these detector settings, we conducted a sensitivity study over a wide range of τ and w, reported in Table X below. Across this sweep **(τ ∈ {80, 90, 95, 98} and w ∈ {1, 4, 8, 16})**, we observe that for all reasonable windowed configurations (w = 4, 8, 16 with τ ∈ [80, 98]), accuracies remain within a narrow 1–2 point band and response lengths stay on the same scale, indicating that our default choice (τ = 95, w = 4) is representative rather than carefully tuned. In contrast, collapsing to token-level entropy (w = 1) or sequence-level entropy (w = length) reliably harms performance and leads to longer, less efficient chains, suggesting that the gains stem from the windowed entropy formulation itself rather than from fine-grained sequence-length tuning.
>
>
> Table X: Sensitivity of high-window-entropy hyperparameters (percentile threshold τ and window size w) on downstream benchmarks.
>
> | Config | Percentile τ | Window size w          | MathVerse-V Acc | MathVerse-V Len | MathVista Acc | MathVista Len | AIME25 Acc | AIME25 Len | MATH500 Acc | MATH500 Len |
> | :----- | :----------- | :--------------------- | :-------------- | :-------------- | :------------ | :------------ | :--------- | :--------- | :---------- | :---------- |
> | A (**default**)                 | 95  | 4                      | **56.5** | *3198.3*  | *74.6* | **1494.5**  | **61.7** | *22618.8* | **95.2** | *2257.7* |
> | B (lower percentile)           | 80  | 4                      | 55.9 | 3300.4  | 73.9 | 1613.4  | 60.5 | 23128.4 | 93.9 | 2315.7 |
> | C (larger window)              | 95  | 16                     | 55.3 | **3148.2**  | 74.1 | 1743.8  | 60.9 | 24432.4 | 94.2 | 2385.5 |
> | D (moderate window)            | 95  | 8                      | *56.3* | 3223.3  | 74.4 | *1533.4*  | *61.5* | **22382.1** | 94.6 | 2278.2 |
> | E (percentile = 90)            | 90  | 4                      | 56.1 | 3356.4  | **74.8** | 1589.2  | 61.2 | 22879.3 | *95.1* | **2221.8** |
> | F (percentile = 98)            | 98  | 4                      | *56.4* | 3248.1  | 74.5 | 1534.2  | 61.4 | 23544.9 | 94.8 | 2323.9 |
> | G (*token-level entropy*, w = 1) | 95  | 1                      | 54.4 | 3438.2  | 73.5 | 1784.1  | 58.7 | 24533.8 | 93.4 | 2435.6 |
> | H (*sequence-level entropy*)   | 95  | length of sequence     | 54.7 | 3343.4  | 73.8 | 1732.9  | 60.2 | 23458.1 | 93.6 | 2349.0 |
>
> Our default setting **(Config A: τ=95, w=4)** is close to the best or tied-for-best in accuracy, without being aggressively tuned, while nearby configurations (e.g., D/E/F) achieve almost identical results. Only more extreme choices that abandon the windowed design—token-level entropy (G, w=1) or sequence-level entropy (H, w = length)—lead to noticeable degradation in both accuracy and length (longer, less efficient chains), which empirically justifies our use of windowed high-entropy as a detection mechanism. We will clarify this distinction in the revised version by explicitly framing τ and w as robust detector settings rather than task-specific hyperparameters, and by pointing readers to Table X for quantitative evidence of insensitivity.
>
>
> ---

---

> > ### Author Response · Authors · 2025-11-18
> > **Response to Reviewer oeN6 (2/2)**
> >
> > ## W2. Larger scales
> >
> > We thank the reviewer for raising this important point. We fully agree that validating AEPO on larger backbone scales (≥13B–70B) would further strengthen the case for its generality. At this stage, our computational budget and GPU availability constrained us to 3B and 7B models, where we already observe consistent gains across diverse multimodal and textual reasoning benchmarks. Since AEPO’s entropy-shaping mechanism is architecture-agnostic and only relies on batch-level statistics, we expect it to transfer naturally to larger foundation models.
> >
> > We acknowledge this as a limitation of the current work and will make it explicit in the paper. If additional resources become available, we plan to extend our study to larger model scales and include those results in a future version. We appreciate the reviewer’s suggestion, which points to a valuable direction for follow-up experiments.
> >
> > ---
> >
> > ## Additional contribution: high-quality multimodal reasoning dataset.
> >
> > We would also like to highlight a contribution that may  overlook in the main discussion: our self-curated multimodal CoT reasoning dataset ARES-SFT-223K. This dataset is constructed with explicit chain-of-thought annotations and difficulty-level labels across diverse multimodal tasks. Beyond supporting the experiments in this paper, we believe this resource is valuable to the community as a reusable benchmark and training corpus for future multimodal reasoning work, especially for studies on adaptive reasoning and difficulty-aware training. We will make the dataset and its difficulty annotations publicly available to facilitate follow-up research.
> >
> > ---

---

> > > ### Author Response · Authors · 2025-11-27
> > > **Response to Reviewer oeN6**
> > >
> > > Dear Reviewer oeN6,
> > >
> > > Did we satisfactorily answer your questions? Would you like us to clarify anything further? Feel free to let us know, many thanks.
> > >
> > > Best regards,
> > >
> > > Authors

---

> > > > ### Author Response · Authors · 2025-11-28
> > > > **Response to Reviewer oeN6**
> > > >
> > > > Dear Reviewer oeN6,
> > > >
> > > > I hope this message finds you well. As the discussion period is **nearing its end**. I wanted to ensure we have addressed all your concerns satisfactorily. If there are any additional points or feedback you'd like us to consider, please let us know. Your insights are invaluable to us, and we're eager to address any remaining issues to improve our work.
> > > >
> > > > Thank you for your time and effort in reviewing our paper.
> > > >
> > > > Best regards,
> > > >
> > > > Authors

---

### Author Response · Authors · 2025-11-25
**General Response**

**Dear Reviewers, ACs, and SACs,**

We deeply appreciate the insightful and constructive comments provided by all reviewers.

---
We are grateful for the reviewers' recognition of **ARES** as a timely and principled solution to the "overthinking" and "under-exploring" issues in Multimodal Large Reasoning Models (MLRMs). We believe ARES provides a rigorous foundation for **entropy-guided adaptive inference**, effectively bridging the gap between uncertainty modeling and efficient reasoning.

Overall, we are encouraged by the reviewers' positive feedback, which highlights:
* **Novel Conceptual Contribution:** The introduction of High Window Entropy (HWE) as a quantitative, interpretable signal for reasoning exploration is innovative and well-motivated (Reviewers `oeN6`, `A67V`).
* **Principled Methodology:** The dual-stage pipeline (AdaCS + AEPO) is theoretically grounded and effectively aligns reasoning length with task difficulty (Reviewers `oeN6`, `eWR6`).
* **Empirical Effectiveness:** The extensive experiments demonstrate consistent improvements and a favorable efficiency-accuracy balance across multiple benchmarks (Reviewers `oeN6`, `gA48`).

* **Clear Presentation and Logical Flow**: The paper is well-written, accurately expressed, and easy to follow, with intuitive visual demonstrations that effectively assist in understanding the method and experiments (Reviewers `eWR6`, `gA48`).

---

To address the reviewers' concerns—particularly regarding fair baselines, hyperparameter robustness, and detailed validation of design choices—we have conducted several additional experiments and analyses during the rebuttal period, including:

* **Fair Comparison with DAPO:** We implemented a **DAPO-only baseline** under the identical training protocol to isolate the specific gains of our hierarchical entropy reward, demonstrating ARES's superiority in both accuracy and length allocation (Reviewer `eWR6`).
* **Hyperparameter Sensitivity & Robustness:** We conducted sensitivity analyses on window size ($w$) and entropy thresholds. Results show that ARES remains stable across diverse window sizes and entropy percentile thresholds, validating our design choices (Reviewers `oeN6`, `A67V`).
* **Granular Analysis of Adaptation:** We added per-difficulty accuracy/length curves and "trigger usage" statistics. These results explicitly demonstrate that ARES allocates longer reasoning chains to harder problems and shorter ones to easier tasks, validating the "rate-to-length" strategy (Reviewer `eWR6`).
* **HWE vs. Token Entropy:** We provided a direct quantitative comparison between **Window-based Entropy (HWE)** and single-token entropy on major benchmarks, confirming that HWE is a more robust signal for identifying reasoning-critical steps (Reviewer `gA48`).
* **Training Efficiency Analysis:** We reported the training overhead (time/GPU-hours) of ARES relative to standard SFT and RL baselines to address concerns about the cost of the two-stage pipeline (Reviewer `eWR6`).

---

**Summary of revisions:**
* **Clarified Definitions:** Rewrote parts of `Section 3` to rigorously define mathematical symbols upon first use and moved critical implementation details (e.g., branching strategy) from the Appendix to the Main Text for better readability (Reviewer `A67V`).
* **New Results & Analysis:** Added the DAPO comparison, training cost analysis, and response-length metrics for ablations in `Section 4.3` and `Table 2` (Reviewer `eWR6`).
* **Expanded Related Work:** Updated `Section 5` to discuss recent works on adaptive Chain-of-Thought and uncertainty quantification (Reviewer `gA48`).
* **Updated Visualizations:** Refined `Figure 3` and added new plots in the Appendix to visualize the "entropy-difficulty interaction" with a larger sample size (Reviewers `A67V`, `eWR6`).

---

**Additional Contribution:**

**Open-Sourcing ARES-SFT-223K**: Beyond the methodological contributions, we emphasize the value of our self-curated dataset, **ARES-SFT-223K**, constructed with explicit Chain-of-Thought annotations and difficulty-level labels across diverse multimodal tasks. We believe this resource will serve as a rigorous benchmark for future research on adaptive reasoning and difficulty-aware training. We commit to releasing this dataset along with the training code to facilitate reproducibility and community adoption.

---

We sincerely appreciate the reviewers' constructive suggestions and remain committed to continually improving our work. We were particularly encouraged to see that our detailed rebuttal and added experiments have successfully addressed key concerns, leading Reviewer `eWR6` and Reviewer `gA48` to express that **their issues are resolved** and to **raise their rating** for the paper. To elaborate, we address each reviewer's comments point by point below. We thank the reviewers and ACs again for helping us strengthen these aspects.


Best regards,

The Authors

---

### Meta-Review · Area_Chair_FXsr · 2025-12-21

**Summary:**

The recommendation of acceptance is mainly based on the fact that most reviewer concerns are well addressed with added experiments on window size, hyperparameter sensitivity, etc. Two reviewers who originally rated 4 both explicitly mentioned raising the score in discussions. The AC agrees with the reviewers and thinks most of the key concerns and questions are well addressed in rebuttal. I would encourage the authors to incorporate all promised changes and new results in camera ready. See detailed decision rationale below.

**Reviewer Concerns:**

***Reviewers' concerns that were addressed by the rebuttal:***

Hyperparameter Sensitivity & Robustness, fair comparison with baselines, Training Efficiency Analysis, Choice of Window Size etc.

***Reviewers' concerns that are still outstanding:***

Some notations and discussions are to be fixed/added in camera ready; Weak Theoretical Support partially addressed; the scalability of the proposed method

See details below.

**Reviewer Scores:**

***Reviewer oeN6 is likely to maintain the original rating of 6 as the major question is well addressed in rebuttal.***

1. Hyperparameter Sensitivity

Well addressed

2. The scalability of the entropy-shaping mechanism to very large foundation models remains uncertain.

Partially addressed

***Reviewer eWR6 is likely to raise the rating from 4 to 6, as most concerns have been well addressed by the rebuttal, and the reviewer explicitly said "will raise my rating based on these improvements."***

1. unclear whether the gains on downstream reasoning benchmarks are sensitive to these hyperparameters or to sequence length

Mostly addressed

2. Training cost comparison lack

Well addressed

3. Fair comparison needed on on DAPO

Addressed

4. running an RL stage starting from ARES-CS-Vanilla and compare it directly to ARES-RL-7B to verify whether the strategy is actually effective.

Addressed

5. add response-length metrics and curves to the table/analysis.

Addressed

6. include targeted analyses (e.g., per-difficulty accuracy/length and trigger usage) to directly demonstrate adaptation rather than relying only on aggregate length differences.

Addressed

***Reviewer A67V is likely to either maintain or raise the score from 4 to 6 as most of the important question are answered and addressed.***

1. Limited Scale of Empirical Analysis

Addressed

2. Weak Theoretical Support

Mostly addressed

3. Clarity and Readability Issues

Partially addressed

4. Implementation Details of the Exploration Mechanism

Addressed

5. Choice of Window Size

Well addressed

***Reviewer gA48 is likely to raise the score from 4 to 6 as most of the important question are answered and well addressed. The reviewer also explicitly mentions "raise my rating." ***

1. Lack of quantitative validation of window entropy

Well addressed

2. fundamental concept remains closely related to a substantial body of prior work on entropy-based uncertainty estimation

Mostly addressed

3. performance decreases on MathVista. What is the underlying reason for this degradation?

Addressed

---

### Decision · Program_Chairs · 2026-01-26

Accept (Poster)